# Task Characteristic Contexts for Improving Generalization in Offline Meta-Reinforcement Learning

## Abstract

Context-based offline meta-reinforcement learning (meta-RL) methods typically extract contexts summarizing task information from historical trajectories to achieve adaptation to unseen target tasks. Nevertheless, previous methods are affected by context shift caused by the mismatch between the behavior policy and context-based policy, as well as the distinctness among tasks, leading to poor generalization and limited adaptation. Our key insight is that existing methods overlook the task characteristic information, which not only reflects task-specific information but also serves to distinguish among tasks, thereby hindering the extraction and utilization of contexts during adaptation. To address this issue, we propose a framework called task characteristic contexts for offline meta-RL (TCMRL). We consider that such task characteristic information is directly related to task properties, which consist of both reward functions and transition dynamics, and the interrelations among transitions. More specifically, we design a characteristic metric based on context-based reward and state estimators, which utilize task properties to construct the relationships among contexts extracted from entire trajectories. Moreover, we introduce a cyclic interrelation to obtain the interrelations among transitions within sequential subtrajectories from forward, backward and inverse perspectives. Contexts with task characteristic information provide a comprehensive understanding of each task and implicit relationships among them, enabling effective extraction and utilization of contexts during adaptation. Experiments in meta-environments demonstrate the superiority of TCMRL over existing offline meta-RL methods in generating more generalizable contexts and achieving effective adaptation to unseen target tasks.

## 1 Introduction

Context-based offline meta-reinforcement learning (meta-RL) learns how to extract contexts from a series of training tasks and adapt to new tasks. Specifically, contexts encompass crucial statistical task information, which is derived from historical trajectories and used to guide adaptation. Existing methods (Gao et al., 2023; Li et al., 2021b; Yuan & Lu, 2022; Zhou et al., 2024; Nakhaeinezhadfard et al., 2025) learn to generate and utilize contexts from offline trajectories of meta-training tasks during the meta-training phase to avoid expensive online interactions with real or simulated environments. Subsequently, they collect a few online trajectories from unseen target tasks (meta-testing tasks) and leverage contexts extracted from these trajectories to achieve adaptation during the meta-testing phase.

However, existing methods face the challenge of *context shift* (Wang et al., 2023; Gao et al., 2023), which is closely related to the classical memorization problem in meta-learning (Yin et al., 2020) and the Markov decision process (MDP) ambiguity problem (Li et al., 2020; 2021a). This issue arises from the mismatch between the behavior policy and the context-based policy, as well as the inherent distinctness among tasks. Specifically, the context encoder overfits the offline trajectories of meta-training tasks generated by the behavior policy during meta-training, failing to extract effective contexts from the online trajectories collected by the context-based policy on unseen target tasks during meta-testing. Consequently, these methods generate contexts with poor generalization, resulting in limited adaptation.

Our key observation is that the limited generalization of contexts in existing methods arises from the failure to capture *task characteristic information*, which is a crucial component of task information. Such information not only reflects the information of individual tasks but also distinguishes tasks from one

Table 1: Comparison of intra-task similarity and inter-task distinctness for contexts generated by our method, ER-TRL (Nakhaeinezhadfard et al., 2025), UNICORN (Li et al., 2024), and GENTLE (Zhou et al., 2024). Both characteristics are measured using cosine similarity, Euclidean distance, and L1 distance.

| Methods | Intra-task similarity | | | Inter-task distinctness | | |
|---|---|---|---|---|---|---|
| | Cosine similarity ($\uparrow$) | Euclidean distance ($\downarrow$) | L1 distance ($\downarrow$) | Cosine similarity ($\downarrow$) | Euclidean distance ($\uparrow$) | L1 distance ($\uparrow$) |
| TCMRL (ours) | **0.9971** | **0.2592** | **0.7710** | **0.1323** | **4.8553** | **17.0715** |
| ER-TRL | 0.9959 | 0.2653 | 0.8841 | 0.1492 | 4.3222 | 15.2584 |
| UNICORN | 0.9963 | 0.2749 | 0.8654 | 0.1745 | 4.5442 | 16.7882 |
| GENTLE | 0.9885 | 0.3495 | 1.2805 | 0.2195 | 3.8957 | 15.1122 |

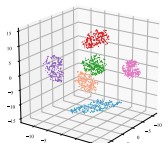 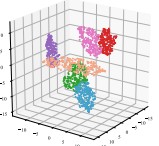

(a) TCMRL (ours)  (b) ER-TRL

Figure 1: 3D t-SNE visualization (van der Maaten & Hinton, 2008) of contexts of TCMRL and ER-TRL on 6 randomly sampled tasks in the Half-Cheetah-Vel environment.

another, thereby constructing intra-task similarity and inter-task distinctness of contexts. The comparison results in Table 1 and Figure 1 demonstrate that existing methods fail to capture these characteristics effectively. Specifically, task characteristic information is directly related to the underlying task properties, which consist of both reward functions and transition dynamics, as well as the interrelations among transitions. First, each transition within the historical trajectories used for extracting contexts is associated with these task properties, and their combination reveals the task characteristic information. Different trajectories of the same task reflect similar task characteristic information, while those from different tasks show clear distinctness. Consequently, the contexts extracted from these trajectories should capture such intra-task similarity and inter-task distinctness, thereby enhancing their generalization. We aim to explicitly model the reward functions and transition dynamics, and construct context relationships based on these task properties, capturing the task characteristic information at the trajectory level. Second, we observe that the interrelations among transitions, revealed by the temporal relationships among transitions, also reflect the task characteristic information. Identifying and exploiting these relationships during meta-training, rather than merely treating transitions as a collection reflecting task reward functions and transition dynamics, captures comprehensive task characteristic information. Consequently, such a comprehensive understanding of tasks leads to more generalizable contexts extracted from sequential trajectories of unseen target tasks, and further facilitates effective adaptation during meta-testing. However, existing methods neglect these interrelations when aggregating the transition representations into contexts, limiting their adaptation performance. Taking into account the limitation of offline trajectories, we capture this critical aspect of task characteristic information at a finer granularity through the subtrajectory level.

To this end, we propose a framework called task characteristic contexts for offline meta-RL (TCMRL) to enhance the generalization of contexts and achieve effective adaptation to unseen target tasks. Specifically, we introduce context-based reward and state estimators to respectively model the reward functions and transition dynamics of tasks, and design a characteristic metric based on these estimations. This metric aims to construct the relationships among contexts based on reward functions and transition dynamics, capturing the task characteristic information at the trajectory level. Additionally, we discover the overlooked interrelations among transitions, which are revealed by the temporal relationships within subtrajectories. For each subtrajectory, we formulate temporal prediction objectives from both forward and backward perspectives, and introduce an inverse model to perform inverse prediction between the first and last transition representations. Then, using subtrajectories as the basic unit, these interrelations are extended to the entire trajectory. All these objectives form our cyclic interrelation, which captures the task characteristic information at the subtrajectory level. Overall, with comprehensive task characteristic information, TCMRL generates generalizable contexts with intra-task similarity and inter-task distinctness, enabling effective adaptation to unseen target tasks. The main contributions of TCMRL are fourfold:

- We propose TCMRL to capture the task characteristic information for mitigating the negative impacts of context shift, generating contexts with generalization, and achieving effective adaptation to unseen target tasks.
- We design a characteristic metric that constructs the relationships among contexts based on task reward functions and transition dynamics, capturing task characteristic information at the trajectory level.
- We introduce a cyclic interrelation to discover the interrelations among transitions from forward, backward and inverse perspectives, capturing task characteristic information at the subtrajectory level.

- Experimental results on meta-environments demonstrate significant performance improvements compared with previous offline meta-RL methods, validating the effectiveness of TCMRL.

## 2 RELATED WORK

**Meta-reinforcement learning.** Meta-reinforcement learning aims to acquire learning strategies from a series of meta-training tasks and achieve adaptation to unseen target tasks. Previous meta-RL studies can be primarily categorized into two distinct methods: context-based methods and optimization-based methods. Context-based methods encode contexts from the critical statistical information about tasks, which is generally presented in the form of historical trajectories. This process is commonly accompanied by the utilization of recurrent (Fakoor et al., 2020; Wang et al., 2017), recursive (Mishra et al., 2018), or probabilistic (Rakelly et al., 2019; Zintgraf et al., 2020) structures. Moreover, optimization-based methods (Finn et al., 2017; Foerster et al., 2018; Houthooft et al., 2018) formalize the process of the task adaptation as the execution of policy gradients over limited samples, aiming to acquire an optimal initialization of the policy. TCMRL is most closely related to the context-based meta-RL.

**Context-based offline meta-reinforcement learning.** Although context-based online meta-RL methods such as SimBelief (Zhang et al., 2025) exist, our work targets the offline setting, where generalizable contexts are derived from offline trajectories rather than from online interactions during meta-training. It aims to adapt to unseen target tasks during the meta-testing phase. FOCAL (Li et al., 2021b) utilizes behavior regularization to restrict the task inference. CORRO (Yuan & Lu, 2022) improves the generalization of contexts through contrastive learning. IDAQ (Wang et al., 2023) leverages a return-based uncertainty quantification to ensure in-distribution contexts of tasks. CSRO (Gao et al., 2023) designs a max-min mutual information representation learning mechanism to reduce the impact of context shift. GENTLE (Zhou et al., 2024) and UNICORN (Li et al., 2024) aim to reconstruct the task models to deepen task understanding. ER-TRL (Nakhaeinezhadfard et al., 2025) reduces the mutual information between task representations and the behavior policy by maximizing the conditional entropy of the policy. However, all these methods fail to capture comprehensive task characteristic information when generating contexts. They rely solely on coarse task labels to construct the relationships among contexts, rather than leveraging the underlying differences among tasks, which lie in both reward functions and transition dynamics. Moreover, they overlook the interrelations among transitions, considering only the task properties reflected in each transition. In contrast, TCMRL uses task properties to establish comprehensive relationships among contexts. Furthermore, it identifies and exploits the interrelations among transitions to capture task characteristic information for handling similar structures during meta-testing. With comprehensive task characteristic information, TCMRL achieves effective adaptation to unseen target tasks.

## 3 PRELIMINARIES

**Reinforcement learning.** The formulation of a reinforcement learning (RL) task commonly takes the form of a fully observable Markov decision process (MDP), which can be defined as a tuple $M = \langle \mathcal{S}, \mathcal{A}, p, r, \gamma, \rho_0 \rangle$. $\mathcal{S}$ is the state space, $\mathcal{A}$ is the action space, $s^t \in \mathcal{S}$ and $a^t \in \mathcal{A}$ respectively represent the state and action at time-step $t$, $p(s^{t+1}|s^t, a^t)$ is the transition dynamics, $r(s^t, a^t)$ is the reward function, $\rho_0$ is the initial state distribution, and $\gamma \in [0,1)$ is the discount factor for future rewards. A stochastic policy is a distribution $\pi(a^t|s^t)$ of actions. Moreover, the definition of the marginal state distribution at time-step $t$ is $\mu_\pi^t(s^t)$ and the primary goal of the agent is to maximize the objective function $max_\pi \mathcal{J}_\mathcal{M}(\pi) = \mathbb{E}_{s^t \sim \mu_\pi^t, a^t \sim \pi}[\sum_{t=0}^{\infty} \gamma^t r(s^t, a^t)]$, which represents the expectation of the accumulated rewards over time.

**Context-based offline meta-reinforcement learning.** Context-based offline meta-RL is generally formalized as partially observable Markov decision processes (POMDPs) (Kaelbling et al., 1998), where states obtained from environments remain only partially visible. It assumes that the information of each task is the unobservable part called the *context* and the agent needs to collect it from offline data as one of the conditions to make decisions: $a_i^t \sim \pi(a_i^t|s_i^t, c_i)$, where $c_i$ is the context related to the task information of task $\mathcal{T}_i$ and the complete state is formed by combining $s_i^t$ and $c_i$. The goal of the meta-agent remains consistent with that of the RL agent. Moreover, meta-RL assumes access to a set of $n_{task}$ meta-training tasks $\mathbb{T} = \{\mathcal{T}_1, ..., \mathcal{T}_{n_{task}}\}$ and a set of unseen target tasks $\mathbb{T}^*$. These tasks differ in their reward functions and/or transition dynamics, and each task is individually modeled as a POMDP. The set of offline datasets

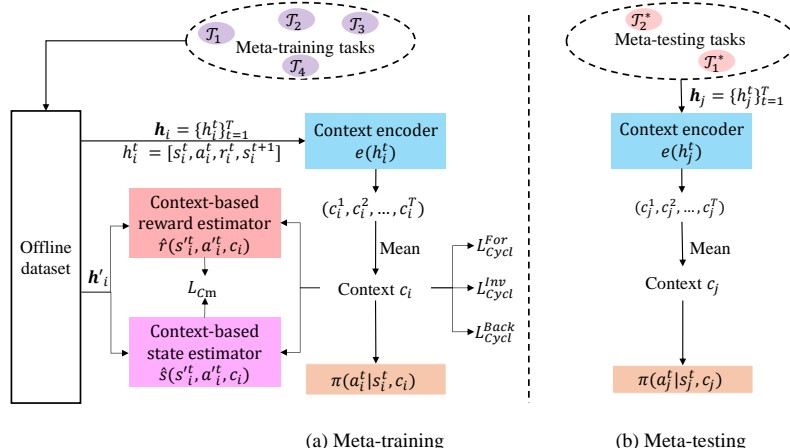

Figure 2: **Framework overview. (a) Meta-training** meta-trains a context encoder $e(h_i^t)$, a context-based reward estimator $\hat{r}(s'^t_i, a'^t_i, c_i)$, a context-based state estimator $\hat{s}(s'^t_i, a'^t_i, c_i)$ and a context-based policy $\pi(a_i^t|s_i^t, c_i)$. $\hat{r}(s'^t_i, a'^t_i, c_i)$ and $\hat{s}(s'^t_i, a'^t_i, c_i)$ are used to construct the characteristic metric loss $L_{Cm}$. The cyclic interrelation loss $L_{Cycl}$ discovers interrelations among transitions from forward, backward and inverse perspectives. **(b) Meta-testing** utilizes the meta-trained modules $e(h_j^t)$ and $\pi(a_j^t|s_j^t, c_j)$ for effective adaptation to unseen target tasks with contexts extracted from a few online trajectories collected from them.

$\mathbb{D} = \{\mathcal{D}_1, ..., \mathcal{D}_{n_{task}}\}$ corresponds to the set of meta-training tasks. More preliminaries of meta-learning and context-based offline meta-RL can be found in Appendices C and D, respectively.

## 4 METHOD

As illustrated in Figure 2, TCMRL comprises two main phases: meta-training and meta-testing. Specifically, during the meta-training phase, TCMRL learns how to extract contexts $c_i$ from historical trajectories $h_i$ sampled from the offline dataset $\mathcal{D}_i$ associated with the meta-training task $\mathcal{T}_i$. During the meta-testing phase, trajectories $h_j$ of the unseen target task $\mathcal{T}_j$ are collected to extract the contexts $c_j$. Such contexts are then used to achieve effective adaptation to $\mathcal{T}_j$. Generally, for $h_i = \{h_i^t\}_{t=1}^T$ that consists of $T$ transitions $h_i^t = (s_i^t, a_i^t, r_i^t, s_i^{t+1})$, the process of context extraction is as follows:

$$\{c_i^t\}_{t=1}^T = e(\{h_i^t\}_{t=1}^T), \qquad c_i = mean(\{c_i^t\}_{t=1}^T). \tag{1}$$

TCMRL primarily operates in the meta-training phase to learn how to capture task characteristic information when generating contexts. Specifically, it (1) models reward functions and transition dynamics of tasks using context-based reward and state estimators, and uses them to construct the relationships among contexts based on the characteristic metric at the trajectory level; and (2) introduces the cyclic interrelation to discover the overlooked interrelations among transitions at the subtrajectory level from forward, backward and inverse perspectives.

### 4.1 CHARACTERISTIC METRIC

**Context-based estimators.** Reward functions and transition dynamics, as the essential differences among meta-tasks, constitute critical task properties.

**Assumption 1 (Deterministic task properties)** *For a particular state-action pair $(s_i^t, a_i^t) \in \mathcal{S} \times \mathcal{A}$ of $\mathcal{T}_i$, and corresponding reward function $r_i(s_i^t, a_i^t)$ and transition dynamic $p_i(s_i^{t+1}|s_i^t, a_i^t)$, $p(r_i^t, s_i^{t+1}|s_i^t, a_i^t, r_i(\cdot, \cdot), p_i(\cdot, \cdot)) = \delta((r_i^t = r_i(s_i^t, a_i^t)) \wedge (s_i^{t+1} = p_i(s_i^t, a_i^t)))$, meaning that both the reward and next state of $(s_i^t, a_i^t)$ are deterministically given by task properties of $\mathcal{T}_i$. Notably, $\delta(\cdot)$ denotes the Kronecker delta function, which returns 1 when the specified condition is satisfied and 0 otherwise.*

GENTLE (Zhou et al., 2024) and UNICORN (Li et al., 2024) construct task properties from offline data using neural networks to deepen the understanding of individual tasks. TCMRL further leverages

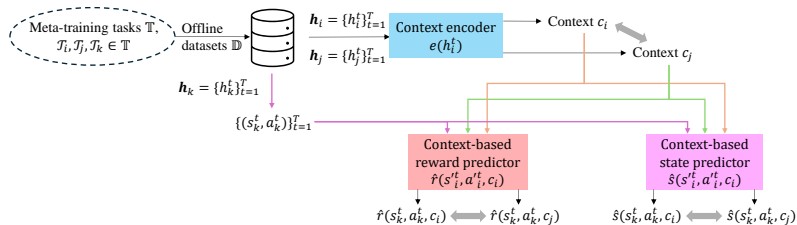

Figure 3: **Characteristic metric.** TCMRL leverages reward functions and transition dynamics to construct relationships among contexts through the characteristic metric. It revolves to $\hat{r}(s'^t_i,a'^t_i,c_i)$, $\hat{s}(s'^t_i,a'^t_i,c_i)$ and trajectories of tasks with label $i$, $j$ and $k$, which may represent three kinds of possible label cases.

reconstructed task properties as constraints to establish both intra-task similarity and inter-task distinctness of contexts, and additionally model the varying degrees of both. Specifically, we introduce context-based reward and state estimators, $\hat{r}(s'^t_i,a'^t_i,c_i)$ and $\hat{s}(s'^t_i,a'^t_i,c_i)$. These two estimators are implemented as learnable neural networks, and are used to construct corresponding task properties. Each estimator receives not only the state $s'^t_i$ and action $a'^t_i$ of the transition $h'^t_i = (s'^t_i,a'^t_i,r'^t_i,s'^{t+1}_i)$ within $\boldsymbol{h}'_i$, but also $c_i$, which encodes task information and enables generalization across all meta-training tasks.

$\hat{r}(s'^t_i,a'^t_i,c_i)$ and $\hat{s}(s'^t_i,a'^t_i,c_i)$ are optimized using supervised objectives to accurately predict the rewards and next states of transitions across all meta-training tasks under corresponding contexts, as follows:

$$L_r = \sum_{t=1}^{T}(\hat{r}(s'^t_i,a'^t_i,c_i)-r'^t_i)^2, \qquad L_s = \sum_{t=1}^{T}(\hat{s}(s'^t_i,a'^t_i,c_i)-s'^{t+1}_i)^2. \qquad (2)$$

Notably, $\hat{r}(s'^t_i,a'^t_i,c_i)$ and $\hat{s}(s'^t_i,a'^t_i,c_i)$ are trained and utilized jointly in an end-to-end manner. Our experimental results show that both $L_r$ and $L_s$ rapidly converge to a low value, indicating that the estimators effectively model these task properties.

**Characteristic metric loss.** As shown in Figure 3, we leverage $\hat{r}(s'^t_i,a'^t_i,c_i)$ and $\hat{s}(s'^t_i,a'^t_i,c_i)$ to impose constraints on relationships among contexts. These constraints, derived from task reward functions and transition dynamics, enforce that the contexts extracted from trajectories capture task-specific information and remain distinguishable. Therefore, the task characteristic information is effectively captured at the trajectory level in a regularized manner and encoded into the contexts to enhance their generalization.

The bisimulation metric (Larsen & Skou, 1991; Ferns et al., 2012; 2011) measures the behavioral similarity between two states based on differences in their rewards and transition dynamics. It ensures that bisimilar states yield identical value functions under a given policy. Given two states $s^\iota$ and $s^\nu$ at time-step $\iota$ and $\nu$ from the same task, an approximate transition dynamics model $\hat{p}: \mathcal{S} \times \mathcal{A} \to M(\mathcal{S}')$, and a policy $\pi$, the bisimulation metric function with approximate dynamics is defined as:

$$d(s^\iota,s^\nu) = c_r|r^\iota-r^\nu|+c_p W_1(d)(\hat{p}(\cdot|s^\iota),\hat{p}(\cdot|s^\nu)), \qquad (3)$$

where

$$r^\iota = \mathbb{E}_{a^\iota \sim \pi}[r(s^\iota,a^\iota)], r^\nu = \mathbb{E}_{a^\nu \sim \pi}[r(s^\nu,a^\nu)], \qquad (4)$$

$W_1(d)$ is the 1-Wasserstein distance, and $c_r \in [0,\infty)$ and $c_p \in [0,1)$ are hyperparameters for weighting.

We are not limited to the transition level but instead build on it to design a characteristic metric that measures the distance between two contexts $c_i$ and $c_j$ through $\hat{r}(s'^t_i,a'^t_i,c_i)$ and $\hat{s}(s'^t_i,a'^t_i,c_i)$. This metric operates on $\boldsymbol{h}_i$ and $\boldsymbol{h}_j$, which are sampled from $D_i$ and $D_j$, the offline datasets of $\mathcal{T}_i$ and $\mathcal{T}_j$, respectively. Both $\boldsymbol{h}_i$ and $\boldsymbol{h}_j$ are encoded into $c_i$ and $c_j$. Then, another trajectory $\boldsymbol{h}_k$ is sampled and used as an anchor to capture task characteristic information. These task labels $i$, $j$ and $k$ associated with $\boldsymbol{h}_i$, $\boldsymbol{h}_j$ and $\boldsymbol{h}_k$ fall into three possible cases: (1) $i=j=k$, where all data is from the same task; (2) two labels are the same, involving two distinct tasks; (3) $i$, $j$ and $k$ are all different, involving three distinct tasks. By accounting for all these cases, the characteristic metric provides a comprehensive measure of context distance through estimations on state–action pairs across tasks, thereby broadening coverage and enhancing generalization. For each $h^t_k = (s^t_k,a^t_k,r^t_k,s^{t+1}_k)$, we apply $\hat{r}(s'^t_k,a'^t_k,c_i)$ and $\hat{s}(s'^t_k,a'^t_k,c_i)$ (and similarly for $c_j$) to predict the reward and next state. Next, by comparing these estimations to the ground-truth $(r^t_k,s^{t+1}_k)$, we measure

**Figure 4: Cyclic interrelation learning** discovers the interrelations among transitions from forward, backward and inverse perspectives. The forward and backward components rely on comparisons across different tasks, while the inverse component focuses on intra-task structure.

the differences under $c_i$ and $c_j$. The definitions of estimation accuracy are as follows:

$$r_i^{acc} = (\hat{r}(s_k^t, a_k^t, c_i) - r_k^t), \qquad r_j^{acc} = (\hat{r}(s_k^t, a_k^t, c_j) - r_k^t), \qquad (5)$$

$$s_i^{acc} = (\hat{s}(s_k^t, a_k^t, c_i) - s_k^{t+1}), \qquad s_j^{acc} = (\hat{s}(s_k^t, a_k^t, c_j) - s_k^{t+1}). \qquad (6)$$

The characteristic metric between $c_i$ and $c_j$ is computed as follows:

$$d_{Cm}(c_i, c_j) = (r_i^{acc} - r_j^{acc}) + (s_i^{acc} - s_j^{acc}). \qquad (7)$$

Additionally, we directly measure the differences between $c_i$ and $c_j$ as follows:

$$d(c_i, c_j) = (c_i - c_j). \qquad (8)$$

Subsequently, we combine these two kinds of differences to formulate our characteristic metric loss $L_{Cm}$. The computation process of $L_{Cm}$ is as follows:

$$L_{Cm} = (d_{Cm}(c_i, c_j)^2 - d(c_i, c_j)^2)^2. \qquad (9)$$

This loss constructs the relationships among contexts according to the estimation accuracy of task reward functions and transition dynamics. Beyond establishing intra-task similarity and inter-task distinctness, it further models the varying degrees of similarity and distinctness, capturing task characteristic information at the trajectory level and enhancing context generalization.

## 4.2 CYCLIC INTERRELATION

As a key component of comprehensive task characteristic information, the interrelations among transitions used to generate the contexts focus on identifying and exploiting the temporal relationships among transitions. These interrelations serve as a complementary means of capturing the task characteristic information beyond task reward functions and transition dynamics. Therefore, learning to extract task characteristic information from this aspect during meta-training enables a more comprehensive understanding of tasks and further facilitates effective adaptation to unseen target tasks when encountering similar structures in the online trajectories during meta-testing. However, existing methods overlook these interrelations and merely treat transitions as independent reflections of task properties, thereby limiting the generalization of contexts. To discover fine-grained interrelations among transitions under limited offline data, we operate on sequential subtrajectories of length $K$ rather than using entire trajectories. Notably, $K$ is a fixed hyperparameter that satisfies $K > 1$. We then design a set of forward, backward, and inverse prediction objectives to construct the cyclic interrelation, which captures comprehensive interrelations among transitions and facilitates the extraction of task characteristic information.

The structure for discovering the interrelations among transitions is illustrated in Figure 4, and further details are provided in Appendix B. Specifically, TCMRL discovers the interrelations within subtrajectories from forward, backward and inverse perspectives. The forward perspective maximizes the mutual information between the prior and last transitions, and the backward perspective does so between the first and subsequent transitions, jointly capturing intra-task similarity and inter-task distinctness. The inverse perspective enhances the task understanding by predicting intermediate transitions from the first and last transitions.

**Forward interrelation learning.** The forward interrelation aligns naturally with the temporal relationships among transitions within subtrajectories. Given a sequential subtrajectory of $\mathcal{T}_i$, we regard the average of transition representations from the first $K-1$ steps as prior context representation $c_{prior}$ and consider the $K$-th step transition representation as target context representation $c_{target}$. Then, we maximize the mutual information between $c_{prior}$ and $c_{target}$:

$$I(m_i^t; c_i^{t+K-1}), \qquad (10)$$

where $m_i^t$ is a convenient representation for $mean(c_i^t, ..., c_i^{t+K-2})$. We approximate the lower bound of the mutual information with the InfoNCE loss function (van den Oord et al., 2018).

Given a sequence of transition representations $\{\{c_i^t\}_{t=1}^T\}_{i=1}^B$ related to $B$ tasks $\{\mathcal{T}_i\}_{i=1}^B$, we operate in two distinct levels of steps. Specifically, we apply the InfoNCE loss to estimate the mutual information in Eq. 10, aiming to capture the interrelations among transitions in the subtrajectories $\{(c_i^t, c_i^{t+1}, ..., c_i^{t+K-1})\}_{i=1}^B$. It relies on the matching relationship between $c_{prior}$ and $c_{target}$, as they encode the same task characteristic information and share the temporal relationships within the subtrajectory. However, discovering forward interrelations from a single subtrajectory is limited, as this process only considers the dependency between transitions within $m_i^t$ and $c_i^{t+K-1}$. Therefore, we extend these interrelations to entire trajectories with sets of subtrajectories as the basic units. This operation allows each transition (except for the first and last $K-1$ steps) to simultaneously contribute to both $m_i^t$ and $c_i^{t+K-1}$, thereby capturing the interrelations between each transition and its surrounding $K$ neighbors. The complete computation process is as follows:

$$L_{Cycl}^{For} = -\frac{1}{T-K+1}\frac{1}{B}\sum_{t=1}^{T-K+1}\sum_{i=1}^{B}log\frac{m_i^t\mathcal{W}c_i^{t+K-1}}{\sum_{l=1}^{B}m_i^t\mathcal{W}c_l^{t+K-1}}, \tag{11}$$

where $\mathcal{W}$ is a learnable parameter that measures the similarity between $m_i^t$ and $c_i^{t+K-1}$, which serve as $c_{prior}$ and $c_{target}$, respectively. Although the computation of $L_{Cycl}^{For}$ in Eq. 11 seems to involve a double loop with time complexity dependent on both $B$ and $T$, it can be efficiently computed through matrix operations, reducing the complexity of the inner loop. Then, the inner loop can be written as follows:

$$\mathcal{L}_{inner} = -\sum_{i=1}^{B}log\frac{m_i^t\mathcal{W}c_i^{t+K-1}}{\sum_{l=1}^{B}m_i^t\mathcal{W}c_l^{t+K-1}} = -\mathrm{Tr}(M), \qquad M_{ij} = log\frac{m_i^t\mathcal{W}c_j^{t+K-1}}{\sum_{l=1}^{B}m_i^t\mathcal{W}c_l^{t+K-1}}. \tag{12}$$

Meanwhile, the outer loop primarily relates to the parallel computations of $c_{prior}$.

**Backward interrelation learning.**    The backward interrelation is similar to the forward one, but serves more as a complementary perspective for discovering interrelations among transitions based on the temporal relationships , thereby capturing the task characteristic information. This perspective operates in the reverse temporal direction, using later transitions to predict earlier ones, thus offering a deeper understanding of the temporal relationships among transitions. Similarly based on mutual information and the InfoNCE loss function, we construct $L_{Cycl}^{Back}$ by treating the average of the last $K-1$ transition representations as $c_{prior}$ and the first transition representation as $c_{target}$. The complete computation process is as follows:

$$L_{Cycl}^{Back} = -\frac{1}{T-K+1}\frac{1}{B}\sum_{t=1}^{T-K+1}\sum_{i=1}^{B}log\frac{m'^t_i\mathcal{W}c_i^t}{\sum_{l=1}^{B}m'^t_i\mathcal{W}c_l^t}, \tag{13}$$

where $m'^t_i = mean(c_i^{t+1}, ..., c_i^{t+K-1})$ and $L_{Cycl}^{Back}$, like $L_{Cycl}^{For}$, can also be computed via matrix operations to reduce computational complexity.

**Inverse interrelation learning.**    The inverse interrelation captures temporal relationships by predicting intermediate transitions within each subtrajectory based on the first and last transitions. We do not impose a strict prediction to match each intermediate transition representation. Instead, the prediction objective is the average of the intermediate transition representations, aligning with the $mean(\cdot)$ operation used in context extraction (Eq. 1). Specifically, we introduce an inverse model $Inv(\cdot, \cdot)$, which takes $c_i^t$ and $c_i^{t+K-1}$ as inputs and predicts $mean(c_i^{t+1}, ..., c_i^{t+K-2})$. The complete computation process is as follows:

$$L_{Cycl}^{Inv} = (Inv(c_i^t, c_i^{t+K-1}) - mean(c_i^{t+1}, ..., c_i^{t+K-2}))^2. \tag{14}$$

Overall, $L_{Cycl}^{For}$, $L_{Cycl}^{Back}$ and $L_{Cycl}^{Inv}$ collectively capture the interrelations embedded in the temporal relationships among transitions within subtrajectories from multiple perspectives, and together constitute our cyclic interrelation loss $L_{Cycl}$. We consider all three perspectives to play important and complementary roles in discovering interrelations among transitions, and assign them equal weight:

$$L_{Cycl} = L_{Cycl}^{For} + L_{Cycl}^{Back} + L_{Cycl}^{Inv}. \tag{15}$$

With $L_{Cycl}$, TCMRL captures comprehensive task characteristic information that includes interrelations among transitions, thereby enhancing the generalization of contexts and enabling effective adaptation to unseen target tasks. Pseudo-codes of both the meta-training and meta-testing phases can be found in Appendix A. The theoretical analysis is in Appendix E and more implementation details are in Appendix H.

## 5    EXPERIMENTS

We evaluate TCMRL on: (1) whether generalizable contexts can be extracted and (2) whether an effective adaptation to unseen target tasks can be achieved.    Our code is available at https://anonymous.4open.science/r/TCMRL-ICLR2026.

Table 2: Comparison results of TCMRL and all baselines in experimental environments.

| Environment | TCMRL (ours) | ER-TRL | UNICORN | GENTLE | IDAQ | CSRO | ANOLE | CORRO | FOCAL |
|---|---|---|---|---|---|---|---|---|---|
| Out-of-Distribution | | | | | | | | | |
| Half-Cheetah-Vel | **-110.54±15.04** | -125.93±7.48 | -124.15±8.47 | -131.01±33.94 | -127.00±21.03 | -126.65±9.13 | -121.77±17.55 | -124.93±24.00 | -144.47±47.94 |
| Point-Robot | **-4.73±0.12** | -4.81±0.14 | -4.78±0.09 | -7.31±1.22 | -4.76±0.07 | -4.78±0.14 | -5.05±0.05 | -5.82±0.50 | -4.96±0.13 |
| Point-Robot-Wind | **-5.55±0.31** | -6.80±1.68 | -15.61±1.15 | -5.98±0.27 | -5.56±0.28 | -15.95±3.36 | -5.81±0.39 | -12.24±4.98 | -5.98±0.27 |
| Sparse-Point-Robot | **12.66±0.24** | 12.31±0.55 | 12.38±1.11 | 5.27±1.16 | 12.45±0.22 | 11.06±1.36 | 11.99±0.93 | 7.22±2.75 | 12.39±0.32 |
| Hopper-Rand-Params | **360.87±15.80** | 345.12±25.95 | 313.87±22.75 | 238.09±21.94 | 314.00±18.59 | 348.78±30.38 | 310.92±49.51 | 256.79±6.48 | 309.70±20.44 |
| Walker-Rand-Params | **328.67±24.48** | 317.30±9.32 | 319.35±14.79 | 320.04±7.35 | 303.43±31.98 | 317.39±19.52 | 315.00±19.64 | 319.01±14.79 | 290.96±31.84 |
| Ant-Goal | **-368.99±5.36** | -616.43±8.17 | -500.71±4.48 | -500.84±2.79 | -417.42±2.20 | -524.41±92.41 | -691.59±4.47 | -614.59±4.10 | -408.26±3.66 |
| Humanoid-Dir | 700.63±21.11 | 558.72±46.12 | 696.56±20.16 | 700.23±19.38 | 582.99±18.47 | 646.90±86.60 | 651.82±49.00 | **738.11±29.83** | 545.19±27.63 |
| In-Distribution | | | | | | | | | |
| Half-Cheetah-Vel | **-109.08±12.79** | -125.55±6.06 | -124.55±6.76 | -132.26±24.00 | -121.29±15.38 | -120.18±14.75 | -120.95±11.24 | -119.26±12.50 | -138.23±13.58 |
| Point-Robot | **-4.71±0.04** | -4.81±0.14 | -4.73±0.02 | -7.63±2.29 | -4.76±0.04 | -4.77±0.05 | -5.11±0.02 | -5.86±0.71 | -4.80±0.02 |
| Point-Robot-Wind | **-5.64±0.10** | -6.63±1.75 | -15.55±1.27 | -5.80±0.16 | -5.69±0.06 | -15.82±3.55 | -5.67±0.18 | -12.13±4.85 | -5.81±0.08 |
| Sparse-Point-Robot | **12.65±0.08** | 12.55±0.19 | 12.29±1.34 | 6.14±0.85 | 12.61±0.09 | 11.18±1.31 | 12.12±0.84 | 5.89±2.07 | 12.62±0.11 |
| Hopper-Rand-Params | **373.58±27.38** | 306.10±24.36 | 286.83±4.49 | 244.51±14.66 | 284.99±6.93 | 286.69±1.01 | 276.64±16.32 | 263.66±12.94 | 279.69±14.80 |
| Walker-Rand-Params | 333.38±22.99 | **342.80±12.50** | 316.30±20.03 | 327.89±17.62 | 306.42±29.85 | 327.68±10.99 | 326.82±36.33 | 338.42±9.22 | 301.76±21.28 |
| Ant-Goal | **-342.64±5.00** | -619.23±8.24 | -517.66±4.13 | -507.70±5.81 | -403.12±6.55 | -490.31±132.81 | -697.59±14.69 | -625.48±8.53 | -392.15±7.04 |
| Humanoid-Dir | 698.20±37.58 | 559.93±50.44 | 697.37±19.56 | 703.53±17.87 | 574.96±15.41 | 645.32±83.16 | 663.06±48.56 | **740.39±28.29** | 547.32±25.02 |

Table 3: Ablation results of TCMRL for analyzing the effects of $L_{Cm}$ and $L_{Cycl}$.

| Environment | TCMRL | TCMRL w/o $L_{Cm}$ | TCMRL w/o $L_{Cycl}$ | $L_{Cycl}$ w/o $L_{Cycl}^{For}$ | $L_{Cycl}$ w/o $L_{Cycl}^{Back}$ | $L_{Cycl}$ w/o $L_{Cycl}^{Inv}$ | $L_{Cycl}$ with $L_{Cycl}^{For}$ | $L_{Cycl}$ with $L_{Cycl}^{Back}$ | $L_{Cycl}$ with $L_{Cycl}^{Inv}$ |
|---|---|---|---|---|---|---|---|---|---|
| Out-of-Distribution | | | | | | | | | |
| Half-Cheetah-Vel | **-110.54±15.04** | -136.39±12.99 | -114.62±13.62 | -112.28±9.79 | -129.33±23.77 | -122.32±9.45 | -115.27±15.22 | -116.94±6.05 | -110.82±6.83 |
| Point-Robot | **-4.73±0.12** | -4.85±0.12 | -4.88±0.07 | -4.76±0.02 | -4.81±0.20 | -4.81±0.11 | -4.75±0.04 | -4.88±0.11 | -4.77±0.16 |
| Point-Robot-Wind | **-5.55±0.31** | -5.64±0.36 | -5.89±0.44 | -5.62±0.43 | -5.73±0.39 | -5.72±0.39 | -5.62±0.49 | -5.60±0.19 | -5.80±0.09 |
| Sparse-Point-Robot | **12.66±0.24** | 10.88±0.37 | 12.22±0.40 | 12.51±0.17 | 12.42±0.32 | 11.58±0.44 | 12.39±0.39 | 11.46±0.35 | 12.44±0.08 |
| Hopper-Rand-Params | **360.87±15.80** | 327.11±23.12 | 352.91±20.08 | 338.60±29.48 | 335.59±11.50 | 355.63±43.55 | 357.60±9.63 | 348.04±14.61 | 347.77±24.09 |
| Walker-Rand-Params | **328.67±24.48** | 311.92±6.36 | 321.83±19.88 | 325.84±9.22 | 323.59±27.68 | 301.76±22.23 | 320.60±23.23 | 316.29±22.16 | 322.18±28.90 |
| Ant-Goal | **-368.99±5.36** | -373.46±4.90 | -393.11±1.97 | -369.38±4.78 | -369.18±6.49 | -369.16±3.96 | -372.75±6.09 | -370.97±3.37 | -408.75±6.13 |
| Humanoid-Dir | **700.63±21.11** | 670.49±33.15 | 655.86±27.63 | 679.37±24.22 | 648.52±11.77 | 676.51±12.54 | 654.75±12.19 | 650.00±29.39 | 656.86±11.38 |
| In-Distribution | | | | | | | | | |
| Half-Cheetah-Vel | **-109.08±12.79** | -140.50±16.15 | -115.42±19.32 | -110.93±8.48 | -129.48±21.13 | -132.54±11.38 | -111.33±10.04 | -119.27±12.43 | -111.37±10.28 |
| Point-Robot | **-4.71±0.04** | -4.79±0.02 | -4.73±0.06 | -4.75±0.05 | -4.74±0.08 | -4.73±0.03 | -4.75±0.04 | -4.72±0.03 | -4.73±0.04 |
| Point-Robot-Wind | **-5.64±0.10** | -5.69±0.10 | -5.66±0.10 | -5.71±0.17 | -5.66±0.15 | -5.72±0.01 | -5.71±0.19 | -5.73±0.10 | -5.65±0.03 |
| Sparse-Point-Robot | **12.65±0.08** | 11.13±0.28 | 12.46±0.15 | 12.55±0.09 | 12.55±0.15 | 12.44±0.04 | 12.55±0.12 | 11.45±0.26 | 12.47±0.14 |
| Hopper-Rand-Params | **373.58±27.38** | 332.81±15.65 | 348.68±6.00 | 338.87±13.97 | 371.32±6.92 | 352.27±18.76 | 367.63±6.37 | 363.37±22.29 | 349.55±15.60 |
| Walker-Rand-Params | **333.38±22.99** | 319.00±14.04 | 328.27±13.87 | 327.13±12.99 | 329.89±29.07 | 330.66±10.74 | 323.12±8.69 | 332.87±24.34 | 332.98±26.25 |
| Ant-Goal | **-342.64±5.00** | -351.58±2.01 | -347.08±5.66 | -348.29±9.20 | -342.84±4.22 | -344.63±2.47 | -349.55±5.53 | -347.69±5.34 | -343.75±4.07 |
| Humanoid-Dir | **698.20±37.58** | 665.19±23.19 | 654.78±27.59 | 674.47±20.58 | 686.42±14.63 | 685.57±17.94 | 652.05±17.30 | 643.56±31.51 | 656.94±12.32 |

**Experimental setup.** We compare TCMRL with ER-TRL (Nakhaeinezhadfard et al., 2025), UNICORN (Li et al., 2024), GENTLE (Zhou et al., 2024), IDAQ (Wang et al., 2023), CSRO (Gao et al., 2023), ANOLE (Ren et al., 2022), CORRO (Yuan & Lu, 2022) and FOCAL (Li et al., 2021b) in the Sparse-Point-Robot, Point-Robot-Wind, Point-Robot, Half-Cheetah-Vel, Hopper-Rand-Params, Walker-Rand-Params, Ant-Goal and Humanoid-Dir environments. Notably, for a fair comparison, we use the same offline datasets for all baselines, which may result in deviations from their originally reported performance. More details about the baselines, the experimental environments and their corresponding datasets are in Appendices G, F and J respectively.

**Comparison with baselines.** We directly compare the performance of TCMRL and the baselines in Table 2. These results demonstrate their adaptation performance during the meta-testing phase, where online trajectories are collected to extract contexts for decision-making. The evaluation encompasses both meta-testing tasks (out-of-distribution) and meta-training tasks (in-distribution). Furthermore, all experimental results are averaged across six random seeds and their variances are measured with a 95% bootstrap confidence interval. Results demonstrate that TCMRL achieves more effective adaptation to unseen target tasks than most baselines, while maintaining strong performance on meta-training tasks. This confirms that TCMRL improves context generalization, and thus adaptation, by leveraging constraints from the characteristic metric and transition interrelations discovered via the cyclic interrelation. Additionally, the complete adaptation processes to unseen target tasks are illustrated in Figure 9 in Appendix I.1.

**Ablation study.** To capture comprehensive task characteristic information, TCMRL employs two main parts: the characteristic metric and cyclic interrelation losses. First, we build two variants of the complete framework: one without the characteristic metric loss (TCMRL w/o $L_{Cm}$) and another without cyclic interrelation (TCMRL w/o $L_{Cycl}$). Second, based on TCMRL, we further investigate the effects of the three components of $L_{Cycl}$: $L_{Cycl}^{For}$, $L_{Cycl}^{Back}$ and $L_{Cycl}^{Inv}$. We construct six additional variants: $L_{Cycl}$ w/o $L_{Cycl}^{For}$, $L_{Cycl}$ w/o $L_{Cycl}^{Back}$, $L_{Cycl}$ w/o $L_{Cycl}^{Inv}$, $L_{Cycl}$ with $L_{Cycl}^{For}$, $L_{Cycl}$ with $L_{Cycl}^{Back}$ and $L_{Cycl}$ with $L_{Cycl}^{Inv}$. Specifically, the former three variants remove one component from $L_{Cycl}$, while the latter use only one component. Results in Table 3 demonstrate that removing any single component degrades the performance of TCMRL for both meta-training and meta-testing tasks. Specifically, removing $L_{Cm}$ weakens the constraints imposed on contexts, while removing $L_{Cycl}$ overlooks the interrelations among transitions. Moreover, the effects of $L_{Cycl}^{For}$, $L_{Cycl}^{Back}$, and $L_{Cycl}^{Inv}$ within $L_{Cycl}$ are more complex. Removing any individual component or using it in isolation leads to a performance drop. However, due to the interactions among these components,

Table 4: Effects of subtrajectory length on interrelations among transitions in TCMRL.

| Environment | $K=2$ | $K=4$ | $K=8$ | $K=16$ | $K=32$ | $K=64$ | $K=128$ |
|---|---|---|---|---|---|---|---|
| | | | | Out-of-Distribution | | | |
| Half-Cheetah-Vel | -133.07±13.12 | -129.80±13.64 | -118.23±32.17 | -117.59±16.61 | **-110.54±15.04** | -114.20±12.99 | -126.32±23.77 |
| Point-Robot | -4.83±0.10 | -4.85±0.21 | -4.84±0.09 | **-4.73±0.12** | -4.75±0.14 | -4.76±0.03 | -4.84±0.13 |
| Point-Robot-Wind | -5.79±0.18 | -5.75±0.44 | -5.78±0.31 | -5.91±0.27 | **-5.55±0.31** | -5.71±0.65 | -5.82±0.31 |
| Sparse-Point-Robot | 12.31±0.41 | 12.49±0.14 | 12.36±0.46 | 12.52±0.27 | **12.66±0.24** | 12.36±0.45 | 12.29±0.20 |
| Hopper-Rand-Params | 328.90±10.82 | 345.49±17.42 | 350.14±16.49 | 351.31±28.58 | 355.59±11.75 | **360.87±15.80** | 317.97±31.07 |
| Walker-Rand-Params | 316.01±25.46 | 319.04±20.84 | **328.67±24.48** | 323.20±18.07 | 322.69±9.21 | 317.18±22.82 | 311.46±14.19 |
| Ant-Goal | -390.03±5.5 | -373.16±0.68 | -373.40±6.64 | -368.99±5.36 | **-366.00±6.65** | -370.91±2.93 | -370.12±1.36 |
| Humanoid-Dir | 684.90±20.38 | **700.63±21.11** | 668.19±42.7 | 649.74±27.26 | 642.62±11.28 | 649.92±27.78 | 607.89±22.11 |
| | | | | In-Distribution | | | |
| Half-Cheetah-Vel | -126.04±9.94 | -120.07±6.33 | -120.07±16.09 | -113.76±14.00 | **-109.08±12.79** | -110.68±3.74 | -125.73±26.39 |
| Point-Robot | -4.87±0.03 | -4.84±0.20 | -4.78±0.04 | **-4.71±0.04** | -4.81±0.03 | -4.76±0.04 | -4.78±0.03 |
| Point-Robot-Wind | -5.74±0.07 | -5.69±0.13 | -5.74±0.07 | -5.67±0.04 | **-5.64±0.10** | -5.71±0.14 | -5.72±0.12 |
| Sparse-Point-Robot | 12.59±0.09 | 12.58±0.07 | 12.35±0.73 | 12.59±0.12 | **12.65±0.08** | 12.58±0.10 | 12.58±0.08 |
| Hopper-Rand-Params | 326.48±20.01 | 355.98±9.22 | 363.01±18.12 | 340.04±14.84 | **387.21±19.13** | 373.58±27.38 | 371.57±16.46 |
| Walker-Rand-Params | 329.31±8.85 | 325.78±23.27 | **333.38±22.99** | 331.28±4.19 | 309.81±9.86 | 303.41±12.95 | 301.53±13.59 |
| Ant-Goal | -341.57±2.46 | -337.97±5.73 | -324.39±3.64 | -342.64±5.00 | -320.04±4.94 | **-319.38±4.85** | -324.76±5.57 |
| Humanoid-Dir | 687.10±24.69 | **698.20±37.58** | 672.81±32.29 | 669.55±28.46 | 672.86±10.23 | 672.13±31.34 | 607.31±25.02 |

(a) TCMRL (ours)     (b) ER-TRL     (c) UNICORN     (d) GENTLE

Figure 5: **t-SNE visualization in Half-Cheetah-Vel** of the learned context vectors of TCMRL, ER-TRL, UNICORN and GENTLE.

using only one does not always result in lower performance than using two. The full combination of $L_{Cycl}^{For}$, $L_{Cycl}^{Back}$, and $L_{Cycl}^{Inv}$ is crucial for effectively discovering the interrelations among transitions.

**Effect of subtrajectory length.** Since TCMRL discovers the interrelations among transitions from subtrajectories, the length $K$ plays a crucial role. We evaluate the impact of $K \in \{2,4,8,16,32,64,128\}$ across meta-environments. Results in Table 4 demonstrate that an appropriate choice of $K$ facilitates effective adaptation, while the optimal value of $K$ may vary across environments. Specifically, both excessively small and overly large values of $K$ lead to suboptimal performance. A small $K$ may capture insufficient interrelations that fail to obtain task characteristic information and introduce ambiguity, while a large $K$ may result in weak interrelations due to overly broad temporal spans. We determine the best $K$ via grid search.

**Visualization analysis.** We report the t-SNE visualization (van der Maaten & Hinton, 2008) of the contexts generated by TCMRL, ER-TRL, UNICORN and GENTLE in the Half-Cheetah-Vel environment. These visualizations include contexts from 10 meta-training and 10 meta-testing tasks randomly sampled from the environment. As shown in Figure 5, ER-TRL, UNICORN, and GENTLE fail to capture comprehensive relationships among contexts, exhibiting poor clustering within individual tasks and significant overlap across different tasks. In contrast, TCMRL reveals both intra-task similarity and inter-task distinctness, demonstrating improved context generalization. Additional t-SNE visualizations in the Hopper-Random-Params environment are shown in Figure 10 in Appendix I.3.

## 6 CONCLUSION

We propose TCMRL, a context-based offline meta-RL method that captures comprehensive task characteristic information at both trajectory and subtrajectory levels. It captures not only the task-specific information of individual tasks but also the implicit relationships among tasks, thereby enhancing the generalization of contexts. With such generalizable contexts, TCMRL achieves effective adaptation to unseen target tasks. Specifically, we design a characteristic metric for constructing the relationships among contexts based on task reward functions and transition dynamics at the trajectory level. Moreover, we introduce a cyclic interrelation to discover overlooked interrelations among transitions within sequential subtrajectories from forward, backward and inverse perspectives. Experiments in deterministic continuous control meta-environments demonstrate the superior performance of TCMRL compared with prior offline meta-RL methods.

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

# A    PSEUDO-CODE

We present the meta-training phase of TCMRL in Algorithm 1 and the meta-testing phase of TCMRL in Algorithm 2. Notably, the data collection process can be divided into two distinct stages. In the initial stage, the agent randomly samples actions $a_j^t$ to collect the trajectory $h_j$ for extracting context $c_j$, while in the subsequent stage, actions $a_j^t$ are sampled based on the context-based policy $\pi(a_j^t|s_j^t, c_j)$.

---

**Algorithm 1** TCMRL meta-training.

---

**Input:** The set of offline datasets $\mathbb{D} = \{\mathcal{D}_i\}_{i=1}^{ntask}$; Context encoder $e(h_i^t)$; Context-based reward estimator $\hat{r}(s'^t_i, a'^t_i, c_i)$; Context-based state estimator $\hat{s}(s'^t_i, a'^t_i, c_i)$; Inverse model $Inv(c_i^t, c_i^{t+K-1})$; Context-based policy $\pi(a_i^t|s_i^t, c_i)$; Q-function $Q$.
  1: **while** not done **do**
  2:     **for** step in training steps **do**
  3:         Randomly select a batch of $B$ tasks $\{\mathcal{T}_i\}_{i=1}^B$;
  4:         Sample historical trajectory $h_i$ from the offline dataset $D_i \sim \mathbb{D}$ corresponding to each $\mathcal{T}_i$;
  5:         Extract $\{c_i^t\}_{t=1}^T$ from $\{h_i^t\}_{t=1}^T$ through $e(h_i^t)$ and generate $c_i$ through $mean(\{c_i^t\}_{t=1}^T)$ for each $h_i$ (Eq. 1);
  6:         Compute $L_r$ and $L_s$ to optimize $\hat{r}(s'^t_i, a'^t_i, c_i)$ and $\hat{s}(s'^t_i, a'^t_i, c_i)$, respectively (Eq. 2);
  7:         Compute $L_{Cm}$ based on $\hat{r}(s'^t_i, a'^t_i, c_i)$ and $\hat{s}(s'^t_i, a'^t_i, c_i)$ (Eq. 5–Eq. 9);
  8:         Compute $L_{Cycl}^{For}$ and $L_{Cycl}^{Back}$ with $\{\{c_i^t\}_{t=1}^T\}_{i=1}^B$ (Eq. 11 and Eq. 13);
  9:         Compute $L_{Cycl}^{Inv}$ with $\{\{c_i^t\}_{t=1}^T\}_{i=1}^B$ and $Inv(c_i^t, c_i^{t+K-1})$ (Eq. 14);
10:         Aggregate $L_{Cycl}^{For}$, $L_{Cycl}^{Back}$ and $L_{Cycl}^{Inv}$ into $L_{Cycl}$ (Eq. 15);
11:         Update $e(h_i^t)$ and $Inv(c_i^t, c_i^{t+K-1})$ to minimize $L_{Cm}$ and $L_{Cycl}$;
12:         Update $\pi(a_i^t|s_i^t, c_i)$ and $Q$ with offline RL algorithm SAC (Haarnoja et al., 2018);
13:     **end for**
14: **end while**

---

---

**Algorithm 2** TCMRL meta-testing.

---

**Input:** The set of unseen target tasks $\mathbb{T}^*$; Context encoder $e(h_j^t)$; Learned context-based policy $\pi(a_j^t|s_j^t, c_j)$; Random explore step $t_r$.
  1: **for** each unseen target task $\mathcal{T}_j \sim \mathbb{T}^*$ **do**
  2:     $h_j = \{\}$;
  3:     **for** $t = 0, ..., T-1$ **do**
  4:         **if** $t < t_r$ **then**
  5:             Agent randomly samples an action $a_j^t$ to collect transition $h_j^t = (s_j^t, a_j^t, r_j^t, s_j^{t+1})$;
  6:         **else**
  7:             Compute context $c_j$ with $e(h_j^t)$ (Eq. 1);
  8:             Agent uses $\pi(a_j^t|s_j^t, c_j)$ to roll out $h_j^t = (s_j^t, a_j^t, r_j^t, s_j^{t+1})$;
  9:         **end if**
10:         $h_j = h_j \cup h_j^t$;
11:     **end for**
12:     Compute context $c_j$ with $e(h_j^t)$;
13:     Roll out $\pi(a_j^t|s_j^t, c_j)$ for evaluation;
14: **end for**

---

# B    STRUCTURE OF PERSPECTIVES OF CYCLIC INTERRELATION

We design a cyclic interrelation loss $L_{Cycl}$ to discover interrelations among transitions within subtrajectories from forward, backward and inverse perspectives. The forward and backward perspectives leverage temporal relationships among transitions and constraints provided by task labels to construct both intra-task similarity and inter-task distinctness based on contrastive learning. The inverse perspective focuses on deepening the understanding within individual tasks rather than conducting cross-task comparisons. Specifically, the forward interrelation loss $L_{Cycl}^{For}$ and backward interrelation loss $L_{Cycl}^{Back}$ are directly computed by constructing different prior contexts $c_{prior}$ and target contexts $c_{target}$, as illustrated in Figure 6 and Figure 7, respectively. In contrast, the inverse interrelation loss $L_{Cycl}^{Inv}$ shown in Figure 8, relies on an inverse model $Inv(\cdot, \cdot)$, which takes the first and last transition representations within the subtrajectory as inputs and predicts the mean of the intermediate transition representations.

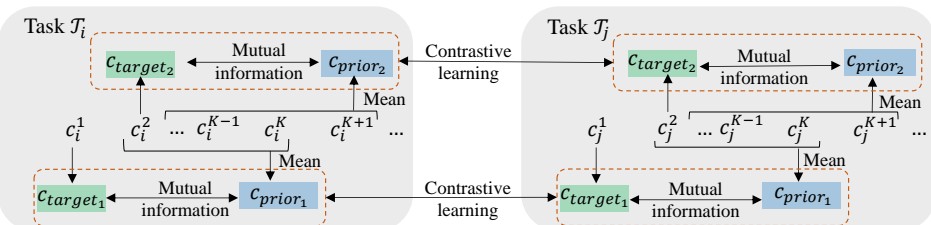

Figure 6: Forward interrelation learning.

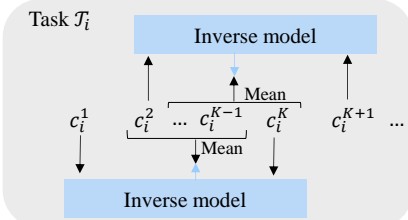

Figure 7: Backward interrelation learning.

Figure 8: Inverse interrelation learning.

## C  PRELIMINARIES OF META-LEARNING

We choose the standard supervised meta-learning to illustrate the concept of meta-learning (see, e.g., (Finn et al., 2017)). We assume tasks $\mathcal{T}_i$ are sampled from a distribution of tasks $p(\mathcal{T})$. The problem setting of the meta-learning consists of two phases: the meta-training phase and the meta-testing phase. These two phases confront distinct sets of tasks, with no overlap between the tasks they encounter. During the meta-training phase, a meta-model is learned through a set of meta-training tasks $\mathbb{T}$. We sample a set of meta-training data $\mathbb{D}$ from these tasks. For a particular task $\mathcal{T}_i$, the corresponding meta-training data $\mathcal{D}_i$ consists of a subset for training $(x_i, y_i)$ and a subset for testing, while $x_i = (x_i^1, x_i^2, ..., x_i^T)$ and $y_i = (y_i^1, y_i^2, ..., y_i^T)$ are sampled from $p(x_i, y_i | \mathcal{T}_i)$, and $x_i^* = (x^{*1}_i, x^{*2}_i, ..., x^{*T}_i)$ and $y^*_i = (y^{*1}_i, y^{*2}_i, ..., y^{*T}_i)$ are sampled from $p(x_i^*, y^*_i | \mathcal{T}_i)$. During the meta-testing phase, the learned meta-model is utilized to address a set of unseen target tasks $\mathbb{T}^*$ and tries to achieve effective adaptation. We denote the meta-parameters learned during the meta-training phase as $\theta$ and the task-specific parameters computed based on the meta-training tasks as $\phi$.

Following Grant et al. (2018) and Gordon et al. (2019), we assess meta-learning algorithms that aim to use the meta-training data $\mathbb{D}$ corresponding to the set of meta-training tasks $\mathbb{T}$ to maximize conditional likelihood $q(\hat{y}^* = y^* | x^*, \theta, \mathbb{D})$, which is related to three distributions: $q(\theta | \mathbb{D})$ that generates the distribution of the meta-parameters $\theta$ from the meta-training data $\mathbb{D}$, $q(\phi | \mathcal{D}_i, \theta)$ that generate the distribution of the task-specific parameters $\phi$ and $q(\hat{y}^* | x^*, \phi, \theta)$ that is the predictive distribution. The learning objective of these distributions is as follows:

$$-\frac{1}{N}\sum_i \mathbb{E}_{q(\theta|\mathbb{D})q(\phi|\mathcal{D}_i,\theta)}\left[\frac{1}{T}\sum_{(x^*,y^*)\in\mathcal{D}_i}\log q(\hat{y}^* = y^*|x^*,\phi,\theta)\right]. \tag{16}$$

Meta-learning algorithms can be primarily categorized into two kinds of distinct algorithms: optimization-based algorithms and context-based algorithms. Specifically, MAML (Finn et al., 2017) is a classic

optimization-based meta-learning algorithm. Within MAML, $\theta$ and $\phi$ denote the weights of the predictor network, $q(\phi|\mathcal{D}_i,\theta)$ is a delta function that is positioned at a location determined through gradient optimization, and $\phi$ parameterizes the predictor network $q(\hat{y}^*|x^*,\phi)$. Moreover, it utilizes the meta-training data $\mathcal{D}_i$ and the parameter $\theta$ in the predictor model for determining the task-specific parameter $\phi$, and this process is as follows:

$$\phi=\theta+\frac{\alpha}{T}\sum_{(x,y)\in\mathcal{D}_i}\nabla_\theta\log q(y|x,\phi=\theta). \tag{17}$$

Meanwhile, the conditional neural processes (CNP) (Garnelo et al., 2018) is a notable context-based algorithm, which defines $q(\phi|\mathbb{D},\theta)$ as a mapping from $\mathbb{D}$ to the parameter $\phi$. Features $e(\mathbb{D})$ extracted from the meta-training data are aggregated through a network $agg_\theta(\cdot)$, and the output is computed through $\phi = agg_\theta \cdot e(\mathbb{D})$. Subsequently, the parameter $\theta$ defines a predictor network that inputs $\phi$ and $x^*$ and outputs the prediction of the distribution $q(\hat{y}^*|x^*,\phi,\theta)$.

## D  PRELIMINARIES OF CONTEXT-BASED OFFLINE META-RL

We assume that context-based offline meta-RL corresponds to a set of tasks consisting of a series of meta-training tasks and a series of meta-testing tasks (unseen target tasks). These tasks within this set shares the same state space $\mathcal{S}$ and action space $\mathcal{A}$, but exhibit variations in their transition dynamics $p(s_i^{t+1}|s_i^t,a_i^t)$ or reward functions $r(s_i^t,a_i^t)$. Moreover, a distribution of these tasks is modeled as joint distribution of transition dynamics $p(s_i^{t+1}|s_i^t,a_i^t)$ and reward functions $r(s_i^t,a_i^t)$, with the following form:

$$\begin{aligned}p(\mathcal{T})&=p(p(s_i^{t+1}|s_i^t,a_i^t),r(s_i^t,a_i^t))\\&=p(p(s_i^{t+1}|s_i^t,a_i^t))p(r(s_i^t,a_i^t)).\end{aligned} \tag{18}$$

This task distribution corresponds to a series of MDPs, and a meta-policy designed by context-based offline meta-RL methods aims to perform well across all these MDPs. These MDPs are formed as POMDPs since they consider the task information of each task to be the unobservable part. Consequently, a context encoder $e(\cdot)$ is utilized to map the task information of the historical trajectory $\boldsymbol{h}$ that corresponds to the task $\mathcal{T}$ to a representation of the context $c\in C$, where $C$ is the space of contexts. The form of the augmented state is as follows:

$$\mathcal{S}_{\text{aug}}\leftarrow\mathcal{S}\times\mathcal{C},\quad s_{\text{aug}}\leftarrow\text{concat}(s,c). \tag{19}$$

This set of MDPs is also defined as task-augmented MDP (TA-MDP) (Li et al., 2021b;a).

Previous context-based offline meta-RL methods (Li et al., 2021b; Rakelly et al., 2019; Wang et al., 2023) typically obtain task information of task $\mathcal{T}_i$ by aggregating transitions from the historical trajectory $\boldsymbol{h}_i^{1:t}=\{s_i^1,a_i^1,r_i^1,s_i^2...,s_i^t,a_i^t,r_i^t,s_i^{t+1}\}$ into a representation of the continuous latent space of contexts $\mathcal{C}$. These methods have proved that the quality of contexts, or the ability of the context encoder to extract task information from historical trajectories, directly influences the performance of the meta-policy and its adaptation to unseen target tasks. In addition, as a traditional and successful context-based offline meta-RL method, probabilistic representations for actor-critic RL (PEARL) (Rakelly et al., 2019) generates contexts $c_i$ in the form of vectors. Moreover, the complete process of adaptation to unseen target tasks involves sampling the vector $c_i$ from the corresponding probabilistic distribution $q_e(c_i|\boldsymbol{h}_i)$, which is parameterized by an encoder $e$. Here, $\boldsymbol{h}_i$ is a complete historical trajectory corresponding to the episode of task $\mathcal{T}_i$. Specifically, the context encoder is implemented by a neural network and the input historical trajectory consists of a series of transitions $h_i^t = (s_i^t,a_i^t,r_i^t,s_i^{t+1})$. Additionally, the context $c_i$ is one of the inputs of the context-based policy $\pi(a_i^t|s_i^t,c_i)$ for making action decisions.

## E  THEORETICAL ANALYSIS

### E.1  CONTEXT-BASED REWARD AND STATE ESTIMATORS

Inspired by UNICORN (Li et al., 2024), we introduce context-based reward and state estimators, $\hat{r}(s'^t_i, a'^t_i, c_i)$ and $\hat{s}(s'^t_i, a'^t_i, c_i)$. They aim to optimize the context-based offline meta-RL through

reconstructing rewards and next states to maximize the mutual information $I(r_i^t,s_i^{t+1}|s_i^t,a_i^t,c_i)$:

$$I(r_i^t,s_i^{t+1};s_i^t,a_i^t,c_i) =$$

$$\int p(h_i^t)p(c_i|h_i^t)\log\frac{p(r_i^t,s_i^{t+1}|s_i^t,a_i^t,c_i)}{p(r_i^t,s_i^{t+1})}$$

$$= \int p(h_i^t)p(c_i|h_i^t)\log p(r_i^t,s_i^{t+1}|s_i^t,a_i^t,c_i)$$

$$\simeq \int_{h_i^t,c_i} \underbrace{q(c_i|h_i^t)}_{encoder}\log \underbrace{p_{\hat{r},\hat{s}}(r_i^t,s_i^{t+1}|s_i^t,a_i^t,c_i)}_{decoder}$$

$$= \mathbb{E}_{e(h_i^t)}\big[\log p_{\hat{r},\hat{s}}(r_i^t,s_i^{t+1}|s_i^t,a_i^t,c_i)\big].$$

### E.2 CHARACTERISTIC METRIC

Our characteristic metric involves three task labels ($i$, $j$ and $k$) and considers the following three scenarios: (1) all task labels are identical, with all data coming from the same task ($i=j=k$); (2) two task labels are identical, involving data from two different tasks; (3) all task labels are distinct ($i\neq j\neq k$). Our $L_{Cm}$ effectively captures task characteristic information and enhances the generalization of contexts across all these scenarios.

Let $D_k=\{h_k^t\}_{t=1}^{T}$ be an offline dataset sampled from task $\mathcal{T}_k$, and $c_i$ and $c_j$ be two contexts, possibly from different tasks. We define the characteristic metric as follows:

$$d_{Cm}(c_i,c_j;D_k) = \mathbb{E}_{(s_k^t,a_k^t)\sim D_k}[((\hat{r}(s_k^t,a_k^t,c_i)-r_k^t)$$

$$-(\hat{r}(s_k^t,a_k^t,c_j)-r_k^t))+((\hat{s}(s_k^t,a_k^t,c_i)-s_k^{t+1})-(\hat{s}(s_k^t,a_k^t,c_j)-s_k^{t+1}))]$$

$$= \mathbb{E}_{(s_k^t,a_k^t)\sim D_k}[(\hat{r}(s_k^t,a_k^t,c_i)-\hat{r}(s_k^t,a_k^t,c_j))+(\hat{s}(s_k^t,a_k^t,c_i)-\hat{s}(s_k^t,a_k^t,c_j))].$$

Then, we design our characteristic metric loss $L_{Cm}$:

$$L_{Cm}=L_{Cm}(c_i,c_j;D_k) = (d(c_i,c_j;D_k)^2-(c_i-c_j)^2)^2.$$

In addition to Assumption 1, we further assume the following:

**Assumption 2 (Decoder expressiveness)** *For any task $\mathcal{T}_i$, there exists context $c_i^*$ such that:*

$$\hat{r}(s,a,c_i^*)=r_i(s,a), \qquad \hat{s}(s,a,c_i^*)=p_i(s,a).$$

**Assumption 3 (Decoder smoothness (Lipschitz))** *There exist constants $l_r$ and $l_s$ s.t.*

$$|\hat{r}(s_k^t,a_k^t,c_i)-\hat{r}(s_k^t,a_k^t,c_j)|\leq l_r\|c_i-c_j\|_2,$$

$$\|\hat{s}(s_k^t,a_k^t,c_i)-\hat{s}(s_k^t,a_k^t,c_j)\|_2\leq l_s\|c_i-c_j\|_2.$$

**Case 1.** In this case, task labels $i=j=k$. From Assumption 1:

$$\hat{r}(s_k^t,a_k^t,c_i)=\hat{r}(s_k^t,a_k^t,c_j)=r_k(s_k^t,a_k^t)=r_k^t,$$

$$\hat{s}(s_k^t,a_k^t,c_i)=\hat{s}(s_k^t,a_k^t,c_j)=p_k(s_k^t,a_k^t)=s_k^{t+1}.$$

It means that $d_{Cm}(c_i,c_j;D_k)=0$. Hence:

$$L_{Cm}=(c_i-c_j)^4.$$

Minimizing $L_{Cm}$ promotes context consistency within the same task, ensuring that different trajectories from the same task yield similar contexts.

**Case 2.** In this case, task labels $i$ and $j$ are different, while $k$ is equal to one of them. Without loss of generality, we consider the example where $k=i\neq j$. We define the estimation errors:

$$\delta_r^i=\hat{r}(s_k^t,a_k^t,c_i)-r_i(s_k^t,a_k^t),$$

$$\delta_r^j=\hat{r}(s_k^t,a_k^t,c_j)-r_j(s_k^t,a_k^t),$$

$$\delta_s^i=\hat{s}(s_k^t,a_k^t,c_i)-p_i(s_k^t,a_k^t),$$

$$\delta_s^j=\hat{s}(s_k^t,a_k^t,c_j)-p_j(s_k^t,a_k^t).$$

Then:

$$\hat{r}(s_k^t,a_k^t,c_i)-\hat{r}(s_k^t,a_k^t,c_j)$$

$$=\hat{r}(s_k^t,a_k^t,c_i)-(r_i(s_k^t,a_k^t)-r_i(s_k^t,a_k^t)-r_j(s_k^t,a_k^t)+r_j(s_k^t,a_k^t))-\hat{r}(s_k^t,a_k^t,c_i)$$

$$=(\hat{r}(s_k^t,a_k^t,c_i)-r_i(s_k^t,a_k^t))+(r_i(s_k^t,a_k^t)-r_j(s_k^t,a_k^t))-(\hat{r}(s_k^t,a_k^t,c_i)-r_j(s_k^t,a_k^t))$$

$$=(\delta_r^i-\delta_r^j)+(r_i(s_k^t,a_k^t)-r_j(s_k^t,a_k^t))$$

$$\le|r_i(s_k^t,a_k^t)-r_j(s_k^t,a_k^t)|+|\delta_r^i|+|\delta_r^j|.$$

Using Assumption 3:

$$|\delta_r^i|\le l_r\cdot||c_i-c_i^*||_2, |\delta_r^j|\le l_r\cdot||c_j-c_j^*||_2.$$

Similarly, for the term of transition dynamics. Hence, we get:

$$d_{Cm}(c_i,c_j;D_k)\le\mathbb{E}_{(s_k^t,a_k^t)}[|r_i(s_k^t,a_k^t)-r_j(s_k^t,a_k^t)|+||p_i(s_k^t,a_k^t)-p_j(s_k^t,a_k^t)||_2]+\epsilon,$$

$$\epsilon=l_r\cdot||c_i-c_i^*||_2+l_r\cdot||c_j-c_j^*||_2+(l_s\cdot||c_i-c_i^*||_2+l_s\cdot||c_j-c_j^*||_2).$$

Minimizing $L_{Cm}$ aligns context distance with the discrepancy of reward functions and transition dynamics across tasks, so larger reward and dynamics differences across tasks are reflected in larger context distances, thereby encouraging the inter-task distinctness.

**Case 3.** In this case, task labels $i$, $j$ and $k$ are different, $i\neq j\neq k$. Now both $c_i$ and $c_j$ are decoupled from the offline dataset $D_k$. Estimation errors are defined as before. We consider:

$$|\hat{r}(s_k^t,a_k^t,c_i)-\hat{r}(s_k^t,a_k^t,c_j)|=|r_i(s_k^t,a_k^t)-r_j(s_k^t,a_k^t)+\delta_r^i-\delta_r^j|$$

$$\le|r_i(s_k^t,a_k^t)-r_j(s_k^t,a_k^t)|+|\delta_r^i|+|\delta_r^j|,$$

where errors are controlled by:

$$|\delta_r^i|\le l_r\cdot||c_i-c_i^*||, |\delta_r^j|\le l_r\cdot||c_j-c_j^*||.$$

Same for transition dynamics. Thus:

$$d_{Cm}(c_i,c_j;D_k)\le\mathbb{E}_{(s_k^t,a_k^t)}[|r_i(s_k^t,a_k^t)-r_j(s_k^t,a_k^t)|+||p_i(s_k^t,a_k^t)-p_j(s_k^t,a_k^t)||_2]+\epsilon,$$

where $\epsilon$ remains consistent with its definition in Case 2. The minimization condition for the loss $L_{Cm}$ is also the same as in Case 2. This case enables probing the gap in task properties between tasks $\mathcal{T}_i$ and $\mathcal{T}_j$, even without directly using their own data. It demonstrates that $L_{Cm}$ can effectively establish the connection between contexts and task properties, capture task characteristic information, and construct both intra-task similarity and inter-task distinctness among contexts.

# F EXPERIMENTAL ENVIRONMENTS

- **Point-Robot**. The Point-Robot environment involves navigating a point robot in a 2D space. The robot always starts at the fixed position (0,0), and the goal for each task is located on a unit semi-circle centered at the origin. The objective of each task is to guide the robot from its starting point to the assigned goal. The state space is $\mathbb{R}^2$, representing the (x, y) position of the robot. The action space is $[-1,-1]^2$, corresponding to the movement in the x and y directions. The reward function is defined as the negative Euclidean distance between the current position of the robot to the goal.
- **Sparse-Point-Robot**. The Sparse-Point-Robot environment consists of a 2D navigation problem, simulated by the MuJoCo physics simulator and introduced in PEARL (Rakelly et al., 2019). In this environment setting, each task involves guiding the agent from the origin to a specific goal position situated on the unit circle centered at the origin. The non-sparse reward is defined as the negative of the distance between the current location and the goal position of the agent. In the case of a sparse-reward scenario, the reward is set to 0 when the agent is outside a neighborhood surrounding the goal, which is controlled by the goal radius. Conversely, when the agent is inside this neighborhood, it receives a reward of 1 minus the distance at each step, yielding a positive value. We use the sparse-reward scenario.
- **Point-Robot-Wind**. The Point-Robot-Wind environment is a variant of the 2D navigation environment called Point-Robot. In this variant, each task solely differs in their transition dynamics, while sharing the same reward function. Specifically, each task is characterized by a distinct wind, which is uniformly sampled from $[-l,l]^2$. Consequently, whenever the agent takes a step, it undergoes a drift determined by the corresponding wind.
- **Half-Cheetah-Vel**. The Half-Cheetah-Vel environment serves as a multi-task MuJoCo benchmark wherein tasks exhibit variations in their reward functions. Specifically, definitions of these tasks are revolved around the specification of the target velocity of the agent. The distribution of the target velocity follows a uniform distribution denoted as $U[0,v_{max}]$.

- **Hopper-Rand-Params**. The Hopper-Rand-Params environment controls the forward movement of a single-legged robot. Tasks encompass diverse aspects such as body mass, body inertia, joint damping, and friction. Each parameter is determined by the default value multiplied by a coefficient randomly selected from the range $[1.5^{-3}, 1.5^3]$. The state space is $\mathbb{R}^{11}$ and the action space is $[-1,1]^3$. Meanwhile, the reward function comprises forward velocity and bonuses for staying alive and controlling costs.
- **Walker-Rand-Params**. The Walker-Rand-Params environment controls the forward movement of a bipedal robot. Similar to the Hopper-Rand-Params environment, each parameter is determined using the same method. Meanwhile, the reward function mirrors that of Hopper-Rand-Params. The state space encompasses $\mathbb{R}^{17}$, while the action space is $[-1,1]^6$.
- **Ant-Goal**. The Ant-Goal environment involves controlling a quadruped "ant" robot to navigate toward a target location. For each task, the goal is positioned on a circle of radius 2 centered at the origin (0,0). The state space is $\mathbb{R}^{29}$, which includes the position and velocity of the ant, as well as the angle and angular velocity of its 8 joints. The action is $[-1,1]^8$, with each dimension representing the torque applied to a corresponding joint. The reward function is defined as the negative Euclidean distance to the goal, with an additional control cost penalty.
- **Humanoid-Dir**. The Humanoid-Dir environment involves controlling a "humanoid" robot to move in a specified target direction. For each task, the direction is sampled uniformly from the interval $[0, 2\pi]$. The state space is $\mathbb{R}^{376}$, and the action space is $[-1,1]^{17}$. The reward function is defined as the dot product between the velocity and the target direction of the robot, with additional components including a survival bonus and a control cost penalty.

Additionally, in the meta-RL environments we employed, each task is characterized by distinct goals. In the Point-Robot, Sparse-Point-Robot and Half-Cheetah-Vel environments, their task sets both consist of 100 tasks, of which 80 tasks are designated as meta-training tasks and 20 tasks are designated as meta-testing tasks. In the Point-Robot-Wind environment, its task set comprises 50 tasks, wherein 40 tasks are meta-training tasks and 10 tasks are meta-testing tasks. In the Hopper-Rand-Params, Walker-Rand-Params, Ant-Goal and Humanoid-Dir environments, their task sets both consist of 40 tasks, while 30 tasks are meta-training tasks and 10 tasks are meta-testing tasks. Notably, all these MuJoCo environments have MIT licenses. Moreover, more detailed environment settings can be found in the configuration files provided in our code.

## G   BASELINES

- **FOCAL** (Li et al., 2021b). FOCAL introduces behavior regularization to the learned policy framework while utilizing a deterministic context encoder for efficient task inference. Furthermore, it incorporates a novel negative-power distance metric within a bounded context embedding space, enabling gradient propagation that is decoupled from the Bellman backup process. Specifically, it treats all online experiences as effective data for generating contexts.
- **IDAQ** (Wang et al., 2023). IDAQ is a framework that extends the foundations of FOCAL. It leverages a return-based uncertainty quantification to generate context within the in-distribution. Additionally, it utilizes effective task belief inference methods to tackle new tasks.
- **ER-TRL** (Nakhaeinezhadfard et al., 2025). ER-TRL is an algorithm that approximately minimizes the mutual information between the distribution over the task representations and behavior policy by maximizing the entropy of behavior policy conditioned on the task representations. With such optimization, it aims to mitigate the negative effects of context shift.
- **UNICORN** (Li et al., 2024). UNICORN is a context-based meta-RL optimization scheme that drives a unified and generalized task representation learning objective based on the information bottleneck principle. It aims to combat the context shift by seeking better optimality bounds or approximations of the objective.
- **GENTLE** (Zhou et al., 2024). GENTLE is to learn task representations with generalization under data constraints. It leverages a task auto-encoder (TAE), which is an encoder-decoder structure, to reconstruct both the state transitions and rewards, capturing the generative structure of task models.
- **CSRO** (Gao et al., 2023). CSRO is an approach that addresses the context shift problem with only offline datasets by minimizing the influence of policy in context during both the meta-training and meta-test phases. Specifically, a max-min mutual information representation learning mechanism is designed to diminish the impact of the behavior policy on task representations

during the meta-training phase. The non-prior context collection strategy is introduced to reduce the effect of the exploration policy during the meta-testing phase.

- **ANOLE** (Ren et al., 2022). ANOLE is an algorithm designed for few-shot adaptation based on human preferences. It enables the agent to determine the objectives of new tasks by querying a human oracle, which compares preferences between pairs of behavior trajectories. This algorithm relates the problem to the classical problem known as Rényi-Ulam's game (Rényi, 1961) in information theory and introduces an extension of Berlekamp's volume (Berlekamp, 1964), which is a metric used to quantify uncertainty in noisy preference feedback.
- **CORRO** (Yuan & Lu, 2022). CORRO is a context-based meta-RL framework for addressing the change of behavior policies. It aims to learn how to obtain robust task representations through contrastive learning.

Notably, all these baselines have MIT licenses.

## H  IMPLEMENTATION DETAILS

### H.1  OFFLINE DATA COLLECTIONS

To ensure a fair comparison, we follow the same approach as CSRO in generating the offline datasets, which are used during the meta-training phase (see Appendix J). For each training task, we use SAC (Haarnoja et al., 2018) to train an agent and save the corresponding policy at different training stages as the behavior policy. Each saved policy is used to roll out 50 trajectories within the corresponding environment, constructing the offline datasets. This approach is widely employed in offline meta-RL methods (Li et al., 2021b;a; Yuan & Lu, 2022; Wang et al., 2023; Gao et al., 2023; Zhou et al., 2024).

### H.2  EXPERIMENTAL DETAILS

Our experiments are performed on a machine with NVIDIA GeForce RTX 2080 Ti and implemented with PyTorch. TCMRL uses the Adam optimizer (Kingma & Ba, 2015) with a learning rate of $3e-4$ for the policy, Q-network, V-network, context encoder and our module, and $1e-4$ for the dual critic. The batch size is set to 256, and the discount factor is 0.99. Moreover, the context encoder $e(h_i^t)$, context-based reward estimator $\hat{r}(s'^t_i, a'^t_i, c_i)$, context-based state estimator $\hat{s}(s'^t_i, a'^t_i, c_i)$ and inverse model $Inv(c_i^t, c_i^{t+K-1})$ are implemented with multi-layer perceptron (MLP) neural network architectures, where each hidden layer consists of a fully connected layer. The detailed configurations of these neural networks are available in our code.

We train 50000 steps for the Point-Robot environment, 100000 steps for the Point-Robot-Sparse, Point-Robot-Wind, Hopper-Rand-Params, Walker-Rand-Params and Humanoid-Dir environments, 40000 steps for the Half-Cheetah-Vel environment and 200000 steps for the Ant-Goal environment. Moreover, because the hyperparameter $K$ used in discovering interrelations among transitions is crucial, we carefully set it for each environment to ensure optimal performance. We focus on the adaptation performance of TCMRL on unseen target tasks and use it as the criterion for evaluating the effectiveness of different values of $K$. Specifically, we set $K$ to 32 for the Half-Cheetah-Vel, Point-Robot-Wind, Sparse-Point-Robot and Ant-Goal environments, 16 for the Point-Robot environment, 64 for the Hopper-Rand-Params environment, 8 for the Walker-Rand-Params environment, and 4 for the Humanoid-Dir environment.

### H.3  IMPLEMENTATION OF BASELINES

We re-evaluate the baselines on the experimental environments using the official code provided by the corresponding papers.

The hyperparameters of the baselines are mostly adopted from the original papers. Moreover, to ensure a fair comparison, all baselines and TCMRL are trained using the same settings under the same environments.

### H.4  WAYS TO DETERMINE HYPERPARAMETER

The fixed hyperparameter $K$ is crucial for TCMRL. In our experiments and analysis, since $K$ represents the length of subtrajectories within the full trajectory, it must not exceed the length of the complete trajectory. Moreover, the value of $K$ should not be too small, as it may hinder the effectiveness of discovering

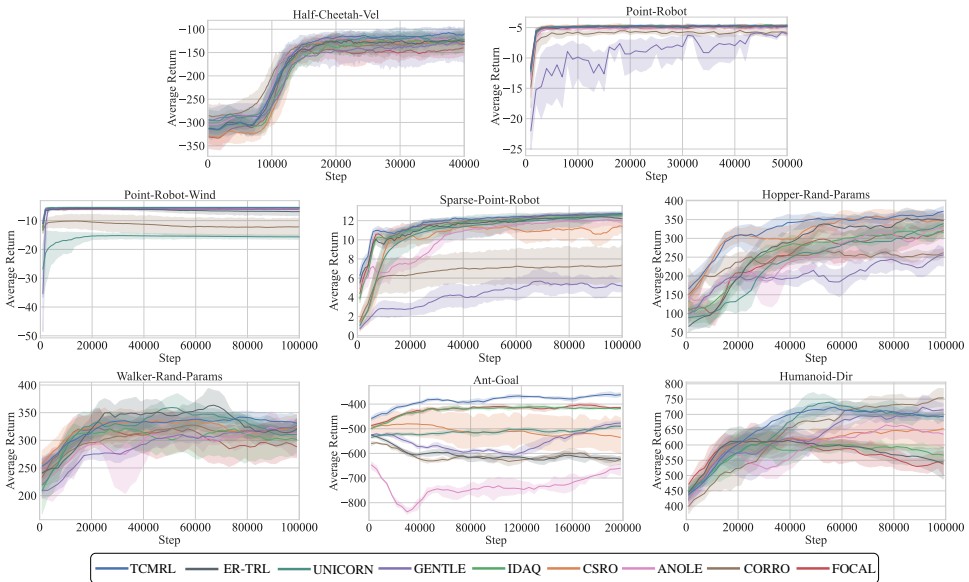

Figure 9: **Comparisons of the adaptation.** The experimental results of TCMRL and baselines in the Half-Cheetah-Vel, Point-Robot, Point-Robot-Wind, Sparse-Point-Robot, Hopper-Rand-Params, Walker-Rand-Params, Ant-Goal and Humanoid-Dir environments.

interrelations among transitions. This hyperparameter may vary across different environments because of the distinctness of these environments. Currently, we determine the optimal $K$ through grid search.

# I ADDITIONAL EXPERIMENTAL RESULTS

## I.1 ADAPTATION PROCESSES

To further compare the adaptation performance of TCMRL and the baselines on unseen target tasks, we present the adaptation processes across all experimental environments in the form of figures. Results in Figure 9 demonstrate that TCMRL achieves more effective adaptation to unseen target tasks than the baselines in most environments. Specifically, in the Point-Robot and Point-Robot-Wind environments, TCMRL and most baselines quickly converge to satisfactory performance. In the Sparse-Point-Robot, Hopper-Rand-Params, Walker-Rand-Params, and Ant-Goal environments, TCMRL starts with relatively high performance and consistently converges to a superior level. In the Half-Cheetah-Vel environment, although TCMRL starts with lower performance, it eventually achieves strong final performance as well. Additionally, in the Walker-Rand-Params environment, the adaptation performance of ER-TRL and UNICORN exhibits significant fluctuations: although they achieve higher rewards than TCMRL during the middle stages, their performance subsequently degrades, eventually converging to a lower level than that of TCMRL. In the Humanoid-Dir environment, TCMRL starts with a relatively low performance but quickly converges to a higher level. Although the final performance is not the best among all methods, it remains competitive.

## I.2 COMPARISON WITH CORRO

We compare TCMRL with CORRO (Yuan & Lu, 2022), a method that generates robust contexts (task representations) through contrastive learning. Specifically, CORRO treats contexts corresponding to the same task as anchor samples and positive samples, respectively, while it constructs negative samples in two different ways. First, in the cases where the overlap of state-action pairs between tasks is larger, it employs a pretrained condition variational auto-encoder (CVAE) (Sohn et al., 2015) for generating negative samples. Second, in the cases where the overlap of state-action pairs between tasks is small, it generates negative samples by reward randomization (RR). The results of CORRO presented in Table 2 represent the maximum performance attained across both CORRO with CVAE and CORRO with RR, serving as a comprehensive result for comparison. The comparative results between TCMRL and CORRO can be

Table 5: Detailed comparison between CORRO and TCMRL.

| Environment | TCMRL (ours) | CORRO (CVAE) | CORRO (RR) |
|---|---|---|---|
| | Out-of-Distribution | | |
| Half-Cheetah-Vel | **-110.54±15.04** | -124.93±24.00 | -131.11±27.26 |
| Point-Robot | **-4.73±0.12** | -14.38±1.89 | -5.82±0.50 |
| Point-Robot-Wind | **-5.55±0.31** | -12.24±4.98 | -12.57±5.67 |
| Sparse-Point-Robot | **12.66±0.24** | 5.23±0.95 | 7.22±2.75 |
| Hopper-Rand-Params | **360.87±15.80** | 256.79±6.48 | 221.66±6.58 |
| Walker-Rand-Params | **328.67±24.48** | 312.78±11.67 | 319.01±14.79 |
| Ant-Goal | **-368.99±5.36** | -638.57±6.95 | -614.59±4.10 |
| Humanoid-Dir | 700.63±21.11 | 713.50±38.82 | **738.11±29.83** |
| | In-Distribution | | |
| Half-Cheetah-Vel | **-109.08±12.79** | -134.54±19.72 | -119.26±12.50 |
| Point-Robot | **-4.71±0.04** | -14.84±0.14 | -5.86±0.71 |
| Point-Robot-Wind | **-5.64±0.10** | -12.13±4.85 | -12.90±5.94 |
| Sparse-Point-Robot | **12.65±0.08** | 5.89±2.07 | 7.63±2.40 |
| Hopper-Rand-Params | **373.58±27.38** | 263.66±12.94 | 248.42±7.80 |
| Walker-Rand-Params | 333.38±22.99 | 316.02±12.36 | **338.42±9.22** |
| Ant-Goal | **-342.64±5.00** | -625.48±8.53 | -627.33±8.82 |
| Humanoid-Dir | 698.20±37.58 | 709.16±37.31 | **740.39±28.29** |

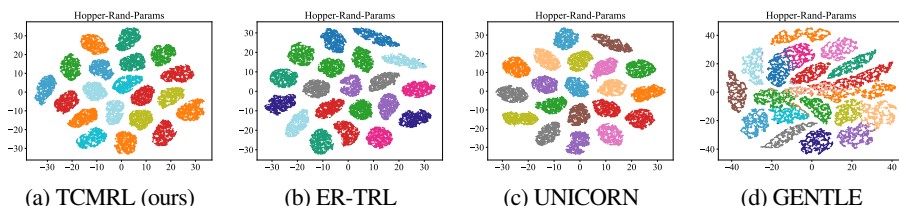

(a) TCMRL (ours)  (b) ER-TRL  (c) UNICORN  (d) GENTLE

Figure 10: **t-SNE visualization in Hopper-Rand-Params** of the learned context vectors of TCMRL, ER-TRL, UNICORN and GENTLE.

found in Table 5. These comparative results demonstrate that TCMRL outperforms both CORRO with CVAE and CORRO with RR across most environments, showcasing superior performance.

### I.3 ADDITIONAL VISUALIZATION ANALYSIS

We present the t-SNE visualization (van der Maaten & Hinton, 2008) of the contexts of TCMRL, ER-TRL, UNICORN and GENTLE in the Hopper-Rand-Params environment. These contexts are related to 10 randomly sampled meta-training tasks and all 10 meta-testing tasks. Visualization results in Figure 10 demonstrate that TCMRL, ER-TRL and UNICORN can effectively construct the comprehensive relationships among contexts of different tasks, reflecting the intra-task similarity and inter-task distinctness. In contrast, the contexts generated by GENTLE fail to form coherent clusters within the same task and exhibit confusion across different tasks, indicating limited generalization.

### I.4 COMPARISON WITH SIMBELIEF

SimBelief (Zhang et al., 2025) is an online meta-RL method that applies the bisimulation metric and operates only coarsely at the trajectory level. In contrast, TCMRL, as a context-based offline meta-RL method, faces a different challenge and performs optimization at both the trajectory and subtrajectory levels. Specifically, TCMRL aims to improve the extraction and utilization of contexts through offline data while avoiding poor adaptation performance on unseen target tasks caused by overfitting. TCMRL leverages the characteristic metric to construct comprehensive relationships among contexts at the trajectory level by capturing intra-task similarity, inter-task distinctness, and varying degrees of both as reflected in task reward functions and transition dynamics. In addition, TCMRL discovers overlooked interrelations among transitions within trajectories to further capture task characteristic information at the subtrajectory

Table 6: Comparison between SimBelief and TCMRL.

| Environment | TCMRL (ours) | SimBelief |
|---|---|---|
| | Out-of-Distribution | |
| Half-Cheetah-Vel | **-110.54±15.04** | -583.51±3.40 |
| Point-Robot | **-4.73±0.12** | -99.56±5.55 |
| Sparse-Point-Robot | **12.66±0.24** | 9.36±2.13 |
| Ant-Goal | **-368.99±5.36** | -889.93±23.70 |
| | In-Distribution | |
| Half-Cheetah-Vel | **-109.08±12.79** | -547.17±4.46 |
| Point-Robot | **-4.71±0.04** | -92.34±5.51 |
| Sparse-Point-Robot | **12.65±0.08** | 6.64±1.93 |
| Ant-Goal | **-342.64±5.00** | -1046.41±29.47 |

Table 7: Loss landscape comparison under the same plane and normalization in the Half-Cheetah-Vel environment.

| Metric | $L_{TCMRL}$ (ours) | $L_{Dm}$ |
|---|---|---|
| Normalized sharpness within $\varepsilon$ (lower is better) | **0.4599** | 0.4744 |
| Normalized Ring-Mean $\Delta L$ (lower is better) | **0.5654** | 0.5788 |
| Hessian condition number at the center (closer to 1 is better) | **1.4145** | 1.6663 |
| Low-loss area fraction (higher is better) | **0.0198** | 0.0177 |

level. By combining these two complementary aspects of optimization, TCMRL captures comprehensive task characteristic information, enhances context generalization and achieves effective adaptation to unseen target tasks. To further compare SimBelief and TCMRL, we apply several experimental settings from TCMRL to SimBelief and conduct experiments in the Half-Cheetah-Vel, Point-Robot, Sparse-Point-Robot, and Ant-Goal environments. Results in Table 6 demonstrate that SimBelief may suffer from poor performance on both meta-training tasks (in-distribution) and unseen target tasks (out-of-distribution) due to limited online learning steps, whereas TCMRL may be constrained by the returns of the offline datasets. Overall, under the same training steps, TCMRL achieves better adaptation performance on unseen target tasks and retains strong performance on meta-training tasks.

### I.5 DISCUSSION ON FLAT MINIMA

To comprehensively explore the effectiveness of our designed $L_{Cm}$ and $L_{Cycl}$, we analyze them from the perspective of flat minima. Specifically, on a fixed 2D loss plane around parameters $\theta_0$, define $\Delta L(\alpha, \beta) = L(\theta_0 + \alpha d_1 + \beta d_2) - L(\theta_0)$, and normalize by the 95th percentile $p_{95}$ of $\Delta L$: $\Delta L_{\text{norm}} = \Delta L / p_{95}$. Let $r = \sqrt{\alpha^2 + \beta^2}$, $r_{\max} = \max r$, and $\varepsilon = \rho r_{\max}$ (default $\rho = 1/3$). This analysis revolves around a series of metrics.

- **Normalized sharpness within $\varepsilon$ (lower is better)**: worst normalized rise within radius $\varepsilon$.
- **Normalized Ring-Mean $\Delta L$ (lower is better)**: average of $\Delta L_{\text{norm}}$ over concentric rings.
- **Hessian condition number at the center (closer to 1 is better)**: anisotropy of local curvature near the center, quantified by $\kappa$.
- **Low-loss area fraction (higher is better)**: fraction of the plane with $\Delta L_{\text{norm}} \leq \tau$ (default $\tau = 0.1$).

FOCAL (Li et al., 2021b) introduces a distance metric, which is widely adopted in existing context-based offline meta-RL methods (Nakhaeinezhadfard et al., 2025; Li et al., 2024; Wang et al., 2023; Gao et al., 2023), to optimize the context encoder. Similar to general contrastive learning functions, this distance metric constructs relationships among contexts solely based on the task labels, without accounting for the varying degrees of similarity and distinctness. The objective of this distance metric is as follows:

$$L_{Dm} = \mathbf{1}\{i = j\} ||c_i - c_j||_2^2 + \mathbf{1}\{i \neq j\} \frac{\zeta}{||c_i - c_j||_2^2 + \epsilon_0}, \tag{20}$$

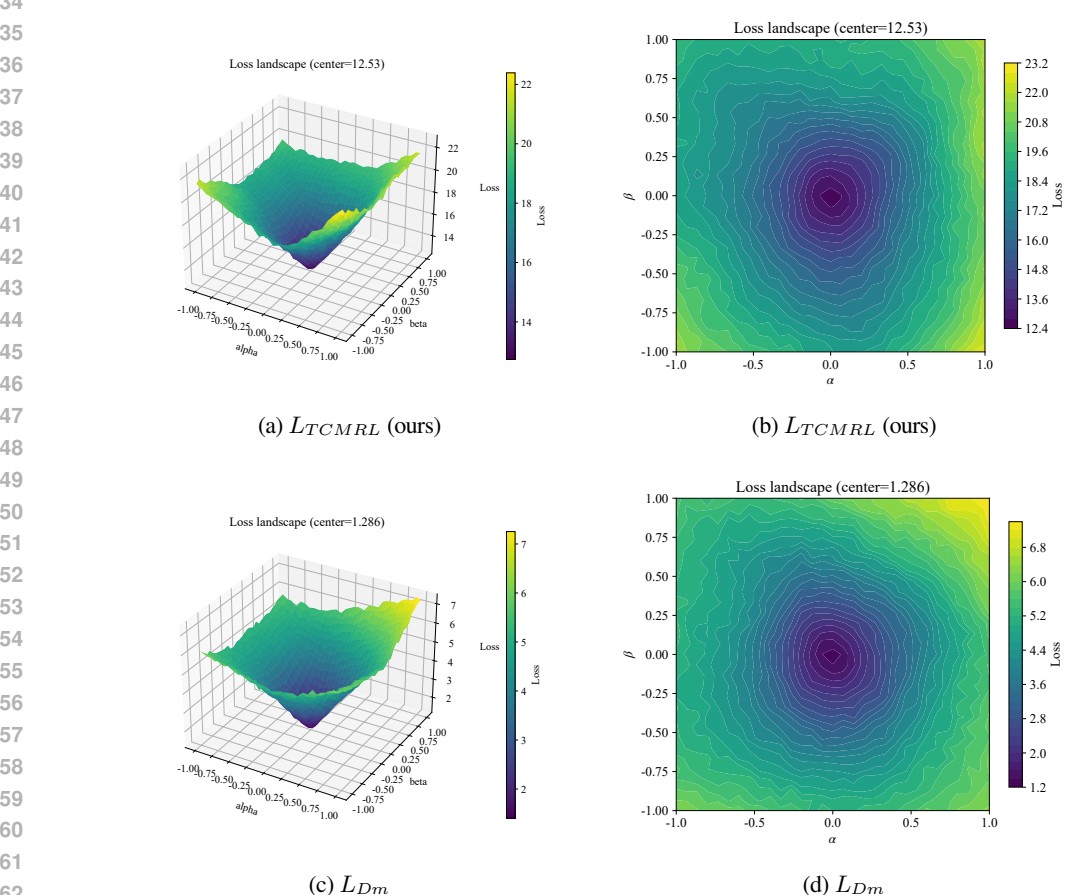

(a) $L_{TCMRL}$ (ours)

(b) $L_{TCMRL}$ (ours)

(c) $L_{Dm}$

(d) $L_{Dm}$

Figure 11: Loss landscape comparison between $L_{Dm}$ and $L_{TCMRL}$ in the Half-Cheetah-Vel environment.

Table 8: Computation cost comparison.

| Method | Testing Time | Training Time | GPU Memory |
|---|---|---|---|
| TCMRL (ours) | 4h12m50s | 2h48m40s | 2107MB |
| TCMRL w/o $L_{Cm}$ | 4h02m33s | 2h35m08s | 1306MB |
| TCMRL w/o $L_{Cycl}$ | 4h09m02s | 2h40m31s | 2104MB |
| ER-TRL | 4h25m14s | 3h28m46s | 1316MB |
| UNICORN | 4h51m33s | 2h41m53s | 1276MB |
| IDAQ | 4h51m25s | 2h35m02s | 1272MB |
| CSRO | 4h02m52s | 2h35m02s | 1274MB |

where $\epsilon_0$ is a hyperparameter introduced to avoid division by zero, and $\zeta$ is a weighting hyperparameter. We compare the loss landscapes of $L_{Dm}$ and $L_{TCMRL}$ under the same plane and normalization in the Half-Cheetah-Vel environment, where $L_{TCMRL} = L_{Cm} + L_{Cycl} + L_{Dm}$, to analyze the auxiliary optimization effects of $L_{Cm}$ and $L_{Cycl}$ relative to using $L_{Dm}$ alone. Results in Table 7 and Figure 11 validate the improved optimization stability and robustness achieved by $L_{Cm}$ and $L_{Cycl}$.

### I.6 COST ANALYSIS OF TCMRL

To assess the computational costs of our proposed TCMRL framework, we experiment in the Half-Cheetah-Vel environment with an RTX 2080 Ti GPU. Following the setup described in Appendix H.2, each experiment consists of a total of 40000 steps. The results in Table 8 demonstrate that the computational costs of TCMRL are manageable and within accepted limits. Moreover, the increased GPU memory

Table 9: Dataset average returns in experimental environments.

| Environment | Dataset Return |
| --- | --- |
| Point Robot | -17.70 |
| Sparse-Point-Robot | 7.24 |
| Half-Cheetah-Vel | -138.29 |
| Point-Robot-Wind | -7.84 |
| Hopper-Rand-Params | 450.84 |
| Walker-Rand-Params | 496.33 |
| Ant-Goal | -379.74 |
| Humanoid-Dir | 737.53 |

usage in TCMRL is primarily attributed to the computation of the characteristic metric loss. We plan to optimize this component in future work. Notably, as an end-to-end framework, we do not compare the computational costs of TCMRL with methods such as GENTLE and CORRO, which require pretraining. This is because pretraining introduces additional and often substantial computational costs.

## J  OFFLINE DATASET RETURNS

Table 9 reports the average returns of the offline datasets, which are utilized in the meta-training phase.

## K  USE OF LARGE LANGUAGE MODELS

We utilize large language models (LLMs) to aid and polish writing.

