# OpenReview forum: "Task Characteristic Contexts for Improving Generalization in Offline Meta-Reinforcement Learning"
_ICLR.cc/2026/Conference — ICLR 2026 Conference Withdrawn Submission_

### Official Review · Reviewer_1Xxc · 2025-10-23

**Soundness:** 2
**Presentation:** 2
**Contribution:** 2
**Rating:** 4
**Confidence:** 5

**Summary:**

This paper proposes a task representation learning framework for context-based offline meta-reinforcement learning, named TCMRL. The main contributions include: (1) a task characteristic metric, and (2) a cyclic interrelation mechanism. The task characteristic metric captures latent task features across different context trajectories, while the cyclic interrelation captures latent features within each individual context trajectory. Experimental results show that TCMRL achieves state-of-the-art or competitive performance across multiple tasks.

**Strengths:**

1. The paper addresses the overlooked issue of extracting task-specific information in existing context-based offline meta-RL (COMRL) methods.
2. The paper provides detailed explanations and presents well-conducted experiments.
3. The proposed method positively contributes to task representation learning in COMRL.

**Weaknesses:**

1. The method figures are difficult to understand without carefully reading the main text. For example, in Figure 2, it is unclear why $h_i'$ is introduced instead of directly using the original $h_i$. The definition and role of $L_{Cm}$ are also confusing; it seems that a reconstruction loss might be more appropriate here. In Figure 3, the calculation process and meaning of the characteristic metric are not clearly explained. In Figure 4, the computation of the three interrelation losses and their specific roles remain unclear.
2. The novelty appears limited. From my perspective, the proposed characteristic metric and interrelation losses mainly aim to increase the latent distance between different tasks and reduce the distance within the same task — effects that are conceptually similar to the contrastive losses used in FOCAL, CORRO, and UNICORN.
3. The paper lacks theoretical justification. There is no theoretical analysis to demonstrate that TCMRL captures more task-relevant information than baseline methods. For instance, CORRO and UNICORN provide mutual-information-based theoretical analyses to support their claims, but this paper does not offer comparable theoretical proof showing that TCMRL achieves tighter bounds or better task information representation.

**Questions:**

1. What advantages do the proposed characteristic metric and interrelation losses offer over the contrastive losses used in FOCAL, CORRO, and UNICORN?
2. Could the authors provide theoretical analysis or proof showing that TCMRL captures more task information than the baseline methods?
3. It would be helpful if the authors could make the method figures more self-contained and easier to understand without extensive reference to the main text.

---

### Official Review · Reviewer_QKQ2 · 2025-10-27

**Soundness:** 2
**Presentation:** 2
**Contribution:** 2
**Rating:** 2
**Confidence:** 5

**Summary:**

The paper proposes to use a characteristic metric based on context-based reward and state estimators to help the learning of the context encoder. Experiment on mujoco environment showcases its edge.

**Strengths:**

1. The paper is easy to follow.
2. The code is given.
3. The ablation study on all the components is thorough.

**Weaknesses:**

1. The paper introduces multiple design components and hyperparameters for training the context encoder. However, the underlying rationale behind these specific choices remains unclear. According to the paper's statement, it appears to build upon the mutual information theory from UNICORN to establish the connection between context representation and the task. However, the proof provided in the appendix seems to somewhat forcibly utilize the Lipschitz assumption to illustrate this connection, lacking further demonstration of why this constitutes an improvement over UNICORN's conclusions.
2. Intuitively, I do think the three possible cases of the Characteristic metric loss essentially mirror the contrastive learning in FOCAL, while the subsequent InfoNCE loss is similar to the approach in CORRO. This appears to be a straightforward combination of UNICORN, FOCAL, and CORRO.
3. According to Equation (7), the reward difference is directly added to the state difference. However, their dimensions are incompatible, making this additive operation mathematically problematic. Furthermore, I find it difficult to justify the direct subtraction in Eq. (2) based on Assumption 1. According to the UNICORN framework, it should be formulated as stocasticity. Why the authors raise this strong assumption is hard to follow.
4. Limited literature: There are some previous works [1] regarding information-theoretic based offline meta RL and some previous works [2] [3] regarding trajectory-level offline meta RL.

[1] Scrutinize What We Ignore: Reining In Task Representation Shift Of Context-Based Offline Meta Reinforcement Learning
[2] Contrabar: Contrastive bayes-adaptive deep rl
[3] Offline Meta Reinforcement Learning -- Identifiability Challenges and Effective Data Collection Strategies

Given the limited novelty, limited intuition, problematic math and limited literature, I lean on rejection for this stage

**Questions:**

1. How do you compute the W_1 distance of the state ?

---

### Official Review · Reviewer_hfoj · 2025-10-29

**Soundness:** 3
**Presentation:** 3
**Contribution:** 2
**Rating:** 4
**Confidence:** 3

**Summary:**

The paper introduces Task Characteristic Contexts for Offline Meta-Reinforcement Learning (TCMRL), aiming to improve context generalization in offline meta-RL. The key idea is to encode task characteristic information (reward and transition dynamics) into context representations using:(1) a characteristic metric that enforces intra-task similarity and inter-task distinctness, and (2) a cyclic interrelation loss that models temporal dependencies among transitions from forward, backward, and inverse perspectives. The authors claim that this helps mitigate context shift and improves adaptation to unseen tasks.  Experiments on multiple benchmarks show that TCMRL outperforms several baselines (ER-TRL, GENTLE, UNICORN, etc.).

**Strengths:**

- Clear problem motivation addressing context shift issue in offline meta-RL.
- Solid experiments on a reasonably wide range of continuous-control meta-RL benchmarks (Half-Cheetah-Vel, Hopper-Rand-Params, Ant-Goal, Humanoid-Dir, …), comparing against several strong baselines (ER-TRL, UNICORN, GENTLE, CSRO, CORRO). Both in-distribution and out-of-distribution evaluations are presented.
- Clear writing and well-structured explanations.

**Weaknesses:**

- The central mechanisms, reward/state prediction for context regularization and multi-directional InfoNCE objectives, are standard techniques already used in contrastive meta-RL (e.g., CORRO, CSRO).

  If I understand correctly, the “characteristic metric” is analogous to bisimulation distance computation, while the “cyclic interrelation“ repurposes bidirectional temporal contrastive learning.

  What is missing is an theoretical analysis showing that combining these elements lyields a fundamentally new capability rather than merely additive regularization.

- The design of the cyclic interrelation loss (L_Cycl) combines several InfoNCE-style objectives across forward, backward, and inverse directions, but the paper provides no theoretical or empirical justification that these directions capture complementary information or yield meaningful gradient diversity.

- The overall framework introduces multiple auxiliary loss terms. However, the paper does not provide a full ablation or efficiency analysis to justify whether this additional complexity is necessary. (The ablation study in Table 3 does not cover all combinations of loss terms.)

- The reported performance gains are modest and often fall within the confidence intervals of the baselines (e.g., ER-TRL, UNICORN).

**Questions:**

See weakness part.

---

### Official Review · Reviewer_L8fY · 2025-10-30

**Soundness:** 3
**Presentation:** 3
**Contribution:** 2
**Rating:** 6
**Confidence:** 2

**Summary:**

The paper introduces Task Characteristic Contexts for Offline Meta-Reinforcement Learning (TCMRL), a framework designed to address the limitations of existing context-based offline meta-RL methods, which suffer from context shift due to policy mismatch and task distinctness, leading to poor generalization and adaptation. TCMRL enhances context generalization by capturing comprehensive task characteristic information at both trajectory and subtrajectory levels. Specifically, it employs a characteristic metric that leverages context-based reward and state estimators to model task properties such as reward functions and transition dynamics, constructing relationships among contexts to ensure intra-task similarity and inter-task distinctness. Additionally, it introduces a cyclic interrelation mechanism that discovers temporal interrelations among transitions from forward, backward, and inverse perspectives within subtrajectories, further enriching task understanding. Experiments across various meta-environments demonstrate that TCMRL outperforms prior methods in generating generalizable contexts and achieving effective adaptation to unseen target tasks.

**Strengths:**

1. The paper is well-written and clearly structured, with a logical flow that effectively explains the method's components and motivations.

2. Empirical results demonstrate that TCMRL outperforms existing models in adaptation and generalization across multiple benchmarks, such as Half-Cheetah-Vel and Hopper-Rand-Params.

3. Comprehensive experiments, including ablation studies and visualizations, provide solid evidence for the method's advantages over prior approaches.

4. The theoretical analysis offers intuitive justification for the characteristic metric, aligning with empirical observations like context clustering in t-SNE plots.

**Weaknesses:**

1. The theoretical analysis is heuristic and descriptive, lacking rigorous formalisms such as generalization bounds or direct validation of loss function designs.

2. The analysis of component interactions in the ablation studies is insufficient. While this does not undermine the core contributions, a deeper investigation would enhance understanding of the method’s inner workings.

3. The paper mentions grid search for key hyperparameters, such as subtrajectory length K, but does not provide an analysis of hyperparameter sensitivity or their impact on performance.

**Questions:**

-

---

### Note · Authors · 2026-01-19

I have read and agree with the venue's withdrawal policy on behalf of myself and my co-authors.